# AutoL2S: Auto Long-Short Reasoning for Efficient Large Language Models

## Abstract

The reasoning-capable large language models (LLMs) demonstrate strong performance in complex reasoning tasks but often suffer from *overthinking* issues after distillation, generating unnecessarily long chain-of-thought (CoT) reasoning paths for easy reasoning questions, thereby increasing inference cost and latency. Recent work largely applies reinforcement learning to shorten reasoning paths in models that already possess reasoning capability. However, these approaches generalize poorly to non-reasoning LLMs, as they assume initial reasoning ability and rely on sparse, outcome-based rewards that make optimization unstable and limit effective learning. In this paper, we propose Auto Long-Short Reasoning (AutoL2S), a dynamic and model-agnostic framework that enables LLMs to adaptively adjust reasoning length according to input complexity, while specifically targeting the stage of transferring non-reasoning LLMs into reasoning-capable but efficient ones via distillation. AutoL2S introduces a learned mechanism in which LLMs are trained on data annotated with long and short CoT paths, together with a special `<EASY>` token that signals when long reasoning can be skipped. During inference, the `<EASY>` token can indicate when the model can skip generating lengthy CoT reasoning. Furthermore, we extend our framework with AutoL2S-Plus, which employs the AutoL2S as a reference model in a length-aware fine-tuning objective to calibrate expected reasoning length, enabling further efficiency gains without loss of accuracy. We theoretically and empirically find that the joint training of long and short CoT paths not only enables dynamic reasoning but also helps the training of shorter CoT generation through knowledge transfer from longer CoT paths. Extensive experiments demonstrate that AutoL2S effectively reduces reasoning length without sacrificing performance, establishing it as an effective framework for scalable and efficient LLM reasoning. The code is available at
`https://anonymous.4open.science/r/AutoL2S-A72E`

## 1 Introduction

Large Language Models (LLMs) have rapidly emerged as essential components in complex reasoning tasks, demonstrating impressive capabilities across advanced applications (Patil, 2025; Chen et al., 2025). However, their deployment in such settings is hampered by fundamental inefficiencies: complex reasoning often requires long-context decoding and extended output generation, which significantly amplifies the computational cost due to the autoregressive nature of LLMs. Specifically, as the reasoning chain grows longer, the computational costs in both memory and inference latency increase quadratically. These issues, known as overthinking problems (Chen et al., 2024; Sui et al., 2025), are further exacerbated by the fact that reasoning-capable LLMs are typically large in scale (Guo et al., 2025), compounding the cost of inference. As a result, practical deployment becomes increasingly cost-prohibitive. This motivates the need for lightweight alternatives that can preserve strong reasoning capabilities while operating with substantially lower resource demands.

To enable scalable deployment of reasoning-capable LLMs, recent works have explored knowledge distillation techniques (Guo et al., 2025; Labs, 2025; Muennighoff et al., 2025; Ye et al., 2025), where non-reasoning LLMs are trained to mimic the reasoning patterns exhibited by stronger reasoning-capable LLMs. These distilled strategies offer significant reductions in parameter and computational cost during training, but simultaneously increase the generated length of chain-of-thought (CoT) reasoning paths, which causes significant cost for decoding. This is because the distillation of-

ten replicates the full long-context reasoning paths from the teacher models (i.e., the LLMs with stronger reasoning capabilities) in order to preserve reasoning performance, resulting in considerable computational overhead during inference. Existing work, such as Qwen3 (Yang et al., 2024a) and Claude (Anthropic, 2023), addresses the overthinking issue by relying on users' manual selection based on prior knowledge to guide LLMs toward either long-form or short-form reasoning. Moreover, other approaches (Luo et al., 2025; Ma et al., 2025a) have primarily employed reinforcement learning (RL) to shorten reasoning paths on top of reasoning-capable LLMs, where these methods heavily rely on models that already exhibit reasoning ability, as these RL-based methods depend on strong reference policies and sparse, outcome-based rewards. However, they still lack the flexibility to dynamically adjust CoT length to the input context, remain constrained by the model's inherent reasoning capability, and are difficult to extend to non-reasoning LLMs.

We identify two fundamental challenges in optimizing towards LLM efficient reasoning. First, the key challenge is to determine when a short reasoning path is sufficient and when a longer one is required. The redundancy of CoT reasoning varies with input complexity, where simple questions can often be answered with minimal reasoning, whereas complex ones demand multi-step reasoning. Without a criterion to adaptively choose between short and long reasoning, models either waste computation on easy cases or risk omitting essential steps for difficult ones. Second, the lack of supervision for short CoT reasoning paths makes it difficult for LLMs to determine the minimal amount of reasoning required to solve a task. Existing training data rarely indicate when shorter reasoning is adequate, making it difficult for non-reasoning models to acquire efficient reasoning ability by learning to omit unnecessary steps without degrading accuracy (Ma et al., 2025b). As a result, even well-aligned LLMs may struggle to identify and retain only the essential reasoning steps (Zhang et al., 2025; Liu et al., 2024; Yu et al., 2024). Therefore, effective CoT compression should be input-aware and dynamically estimate the appropriate reasoning length based on input complexity. However, determining the optimal reasoning length per input is inherently ambiguous, making the efficient-oriented training a challenging and non-trivial problem. This raises a natural question: ***How can we enable LLMs to automatically and dynamically stop overthinking when long and detailed reasoning is unnecessary?***

To address these challenges, we propose Auto Long-Short Reasoning (AutoL2S), a dynamic and model-agnostic framework that enables LLMs to adaptively control reasoning length. AutoL2S automatically determines when long reasoning is necessary and when concise reasoning suffices, thereby bypassing redundant steps without degrading accuracy. Our approach relies on augmented training data that explicitly annotates instances where short CoT reasoning is adequate. This dataset is constructed by pairing long- and short-form CoT reasoning paths, with <EASY> tokens indicating cases where long reasoning can be skipped. We provide both theoretical and mechanistic analyses showing that training with long-form CoT paths improves the quality of short-form reasoning, ensuring robust performance even when the model selects shorter outputs. Furthermore, we extend AutoL2S with a length-aware fine-tuning objective (AutoL2S-Plus) that leverages a reference model to calibrate expected reasoning length, yielding additional compression of CoT without sacrificing correctness. We evaluate AutoL2S on two base LLMs across five reasoning benchmarks spanning mathematics and physics questions. Results show that AutoL2S and AutoL2S-Plus reduces reasoning length by up to 70% while preserving task accuracy. Our contributions are summarized as follows:

- **Auto Long-Short Reasoning.** AutoL2S provides a model-agnostic framework that adaptively selects long or short reasoning paths with <EASY> token based on input complexity.
- **Long2Short Insight.** Theoretical and empirical analyses indicate that long CoT paths benefit the learning of short reasoning, enabling concise outputs without accuracy loss.
- **Extensive AutoL2S-Plus.** We extend AutoL2S as a reference model with a length-aware fine-tuning objective guided to further compress reasoning paths.
- **Reasoning Evaluation.** Across five benchmarks in mathematics and physics, AutoL2S reduces CoT length by up to 70% while preserving performance.

## 2 PRELIMINARY

In this section, we first formally define the Auto Long-Short reasoning problem. We then illustrate the challenges in distilling reasoning capabilities from large reasoning-capable LLMs.

## 2.1 PROBLEM DEFINITION

We study the problem of distilling strong but long reasoning traces from a reasoning-capable LLMs into a smaller, non-reasoning-capable LLMs $f(\cdot \mid \theta)$ with trainable parameters $\theta$, with the goal of enabling it to generate shorter reasoning paths while maintaining task performance. Specifically, let $\mathcal{Y}_{long}$ denote a set of long reasoning responses produced by reasoning-capable LLMs such as DeepSeek-R1 (Guo et al., 2025) or QwQ-32B-preview (Team), and let $\mathcal{Y}_{short}$ denote a set of short reasoning responses from models with inherent shorter reasoning path such as Qwen2.5-Math-7B-Instruct (Yang et al., 2024a). We construct a distillation dataset $\mathcal{D} = \{\mathbf{S}, \mathbf{L}\}$, where $\mathbf{L} \subseteq \mathcal{Y}_{long}$ and $\mathbf{S} \subseteq \mathcal{Y}_{short}$ are collections of valid long and short CoT paths as defined in Definition 1.

The objective is to train $f(\cdot \mid \theta)$ in $\mathcal{D}$ such that, after training, the distilled model $f(\cdot \mid \theta_{\mathcal{D}})$ can automatically adapt its reasoning length to the complexity of the input question, generating a short path when sufficient and a long path when necessary. We expect the outputs of $f(\cdot \mid \theta_{\mathcal{D}})$ to be significantly shorter than those of a model trained only on long-form responses $f(\cdot \mid \theta_{\mathcal{Y}})$, while preserving correctness. This reduction in output length translates directly to fewer generated tokens and thus faster inference. To this end, we propose the *Auto Long-Short Reasoning* framework to enable efficient LLM reasoning through joint utilization of valid long and short CoT paths.

**Definition 1** (**Valid Long and Short CoT Reasoning**). *Let $X$ denote the input with ground-truth answer $y^{\star}$. Let $S = (s_1, \ldots, s_{T_S})$ and $L = (\ell_1, \ldots, \ell_{T_L})$ be token sequences in a vocabulary $\mathcal{V}$, with lengths $T_S \ll T_L$. For $t \in [1, T_S]$ and $t \in [1, T_L]$, we define the prefixes $S_{<t} = (s_1, \ldots, s_{t-1})$ and $L_{<t} = (\ell_1, \ldots, \ell_{t-1})$. We say that $S$ and $L$ are effective short and long CoT paths if every next token is semantically valid given the input and prefix, and the final sequence yields the correct answer $y^{\star}$. Formally, the sets of all such sequences are*

$$\mathbf{S} := \left\{ S \in \mathcal{V}^{T_S} \;\middle|\; (X, S_{<t}) \vdash s_t, \; \forall t \in [1, T_S], \;\; f(X, S) = y^{\star} \right\},$$

$$\mathbf{L} := \left\{ L \in \mathcal{V}^{T_L} \;\middle|\; (X, L_{<t}) \vdash \ell_t, \; \forall t \in [1, T_L], \;\; f(X, L) = y^{\star} \right\}.$$

## 2.2 CHALLENGES OF DISTILLED REASONING LLMS

To equip a non-reasoning model (Qwen (Yang et al., 2024b), Llama3 (Grattafiori et al., 2024)) with reasoning capabilities, DeepSeek-R1 proposes to distill such non-reasoning models using supervised fine-tuning (SFT) with a curated reasoning dataset generated by DeepSeek-R1 (Guo et al., 2025). Rather than relying on reinforcement learning (RL), SFT provides a simpler and more effective approach for enhancing reasoning capabilities. However, SFT-trained reasoning models still face a critical challenge: *they often generate overly lengthy outputs containing redundant or irrelevant content* (Sui et al., 2025). To mitigate this overthinking problem, a series of works leverage SFT to achieve efficient reasoning (Ma et al., 2025a; Xia et al., 2025). Specifically, they curate a reasoning dataset with variable lengths and fine-tune the model on these information-dense samples to develop concise reasoning capabilities.

Striking a balance between brevity and completeness remains non-trivial in removing lengthy outputs. Compressing the reasoning path too aggressively risks omitting essential logical steps, which may degrade model performance on complex tasks. Moreover, in the absence of definitive supervision signals for the minimal sufficient reasoning trace, LLMs may struggle to determine the optimal stopping point for their reasoning process. To address this, we propose the Auto Long-Short Reasoning framework, which encourages LLMs themselves to autonomously decide when to generate shorter or longer reasoning based on the input context.

## 3 AUTO LONG-SHORT REASONING

We systematically introduce the AutoL2S framework in this section. AutoL2S aims to distill reasoning capabilities from reasoning-capable LLMs, allowing the model to learn effective reasoning patterns while reducing the length of reasoning paths required to arrive at correct reasoning answers. In particular, AutoL2S effectively identifies easy questions and applies short reasoning for efficiency, while preserving long-form reasoning only for more complex cases, ultimately resulting in a reduced

average number of generated reasoning tokens. We further present the proposed training methodology and efficient inference process of AutoL2S for efficient LLM reasoning.

### 3.1 TRAINING STAGE OF AUTO LONG-SHORT REASONING

AutoL2S constructs a diverse reasoning dataset by preparing both long and short CoT reasoning paths, based on the complexity of each question. Specifically, long CoT reasoning paths are provided for all questions to capture the complete reasoning process. In contrast, short CoT reasoning paths are more preferable when they can still lead to correct answers, providing more efficient reasoning representation. Formally, questions that are solvable through a short reasoning path are defined as EASY questions. AutoL2S aims to train LLMs not only to learn both long and short reasoning patterns, but also to identify EASY questions, enabling LLMs to perform efficient reasoning when appropriate. More details are provided in Appendix B.

**Constructing Long CoT Reasoning Paths.** We use Bespoke-Stratos-17k (Labs, 2025) as the source of questions. Then, we employ DeepSeek-R1 (Guo et al., 2025) to generate CoT traces along with final answers as the basic long CoT reasoning dataset. Specifically, AutoL2S follows the format in Equation 5 to annotate long CoT reasoning paths and answer for questions in the dataset. The annotation of the reasoning path aims to distill the decision-making capabilities of DeepSeek-R1 into the target model.

**Constructing Short CoT Reasoning Paths for EASY Questions.** Rather than utilizing entire long CoT responses for all questions, an effective reasoning dataset for concise reasoning should contain shorter CoT responses for easy questions. In principle, as long as the answer remains correct, shorter CoT are preferable as training samples. To curate such concise CoT for easier questions, we apply Qwen2.5-Math-7B-Instruct (Yang et al., 2024a) to the same Bespoke-Stratos-17k dataset, generating reasoning traces with shorter CoT trajectories. We employ rejection sampling to filter and retain only those traces that produce correct answers, replacing the corresponding long CoT responses with these shorter alternatives. We annotate the corresponding questions using the <EASY> token. In contrast, questions for which only long CoT responses yield correct answers are without the <EASY> token, and their original long CoT traces are retained in the training dataset.

**AutoL2S Training Strategy.** We distill the reasoning ability from the target model by supervised fine-tuning on the constructed dataset $\mathcal{D}$, which contains paired long- and short-form CoT reasoning paths. Formally, let $f(\cdot|\theta)$ be a targeted non-reasoning base LLM, and $\boldsymbol{x}_i \in \mathcal{D}$ be text data within $\mathcal{D}$. The fine-tuned $f^*(\cdot|\theta_D)$ is optimized using the standard perplexity objective as follows.

$$\min_{\mathcal{D}} \mathcal{L}_{\text{AutoL2S}} = \min_{\mathcal{D}} \mathbb{E}_{\boldsymbol{x}_i \sim \mathcal{D}} \Big[ \frac{1}{|\mathcal{D}|} \sum\nolimits_{i=1}^{|\mathcal{D}|} \log f(x_i \mid x_1, x_2, \cdots, x_{i-1}, \theta) \Big].$$

### 3.2 CONCATENATION ADVANTAGE FOR LONG TO SHORT REASONING TRAINING

In this section, we analyze AutoL2S from both theoretical and empirical perspectives to highlight the mechanism and advantages behind AutoL2S training. We provide Theorem 1 to formalize that concatenating long and short CoT paths benefits the training of the short path, with the improvement quantified by conditional mutual information and mechanism analysis.

**Theorem 1 (Concatenation Advantage for Long–Short CoT Training).** *Let $X$ denote the input, $L = (\ell_1, \ldots, \ell_{T_L})$ the long-CoT token sequence, and $S = (s_1, \ldots, s_{T_S})$ the short-CoT token sequence, with training order $L$ to $S$. Then, the conditional entropy $H(\cdot|\cdot)$ of the next short token satisfies:*

$$H(S_t \mid X, L, S_{<t}) \leq H(S_t \mid X, S_{<t}), \quad \forall t \in [1, T_S]. \tag{1}$$

*Equivalently, averaging across all positions with the improvement quantified as*

$$\frac{1}{T_S} \sum_{t=1}^{T_S} \Big[ H(S_t \mid X, S_{<t}) - H(S_t \mid X, L, S_{<t}) \Big] = \frac{1}{T_S} \sum_{t=1}^{T_S} I(S_t; L \mid X, S_{<t}) \geq 0. \tag{2}$$

*Thus, the long CoT path $L$ provides additional mutual information $I(\cdot|\cdot)$ that strictly increases the entropy of the short CoT path $S$ whenever $L$ is informative about $S$.*

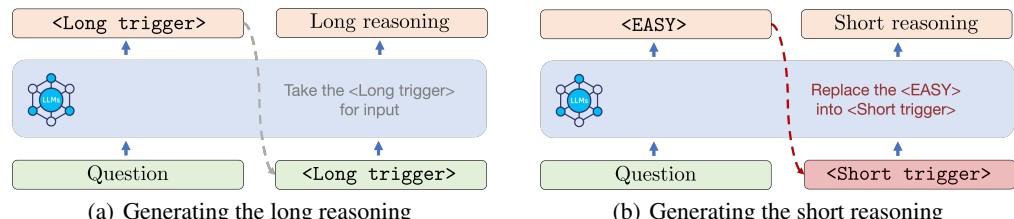

(a) Generating the long reasoning        (b) Generating the short reasoning

Figure 1: During the inference process, LLMs generate (a) a long reasoning path in the case without <EASY> token; and generate (b) a short reasoning path in the case with <EASY> token. Note that the generation of either long or short CoT reasoning paths is automatically determined by the model without any human intervention.

Theorem 1 shows that concatenating the long and short CoT paths reduces the conditional entropy of the short CoT path under long CoT path distillation settings, since the long path provides additional information. In the model training case, where the loss is cross-entropy or equivalently perplexity, the same inequality holds under AutoL2S settings. Thus, long CoT concatenation benefits the training of the short CoT path by providing more information during training. In addition, rejection sampling offers a principled mechanism for aligning the training distribution with our theoretical framework. From Theorem 1, long CoT reasoning paths provide auxiliary information that reduces the entropy of short-path learning; while rejection sampling effectively tunes how much positive signal is preserved in training by filtering short CoT reasoning paths to ensure correctness while varying their informational overlap with long paths.

Importantly, we find that long reasoning paths help LLMs better acquire short reasoning paths. As shown empirically in Section 4.4, providing a long CoT paths supplies additional context that enables the model to learn short CoT reasoning with additional information, where the attention maps become noticeably sparser after training. This offers evidence that long CoT acts as a useful guide for short CoT, consistent with the information-theoretic prediction of Theorem 1.

### 3.3 INFERENCE STAGE OF AUTO LONG-SHORT REASONING

During the inference stage, AutoL2S automatically determines whether to reason with long or short CoT reasoning paths. Specifically, guided by the data formats in Equations (4) and (5), the model begins generation by producing either a <Long Trigger> or an <EASY> token, corresponding to a regular or EASY question, respectively. In practice, as illustrated in Figure 4, AutoL2S dynamically adapts its reasoning strategy based on the initial token generated after receiving a user prompt. If the model outputs a <Long Trigger> token (Figure 4(a)), it indicates that the question requires a long reasoning path; the model then proceeds with standard autoregressive generation to complete the reasoning and produce the final answer. In contrast, if the model generates an <EASY> token (Figure 4(b)), this suggests the question is solvable with a short reasoning path.

Here, Theorem 2 formalizes this mechanism by showing that the choice between long and short CoT paths can be cast as a rule as *risk minimization problem*, balanced by distributional divergence and token cost. Each candidate path is associated with a per-instance risk. The optimal adaptation policy is then obtained by minimizing the expected risk between the two options, which moves to either long or short CoT generation.

**Theorem 2** (**Optimal Adaptation with <EASY> Token**). *Let $p_\theta^L(\cdot \mid X)$ and $p_\theta^S(\cdot \mid X)$ denote the predictive distributions when decoding with the long and short CoT paths $L = (\ell_1, \ldots, \ell_{T_L})$ and $S = (s_1, \ldots, s_{T_S})$, respectively. Given an input $X$, define the per-instance risks as*

$$J_S(X) = \mathbb{E}\Big[ D\Big( p_\theta^S(\cdot \mid X) \,\Big\| \, p_\theta^L(\cdot \mid X) \Big) \Big] \; + \; \lambda \, \mathbb{E}[T_S(X)],$$

$$J_L(X) = \lambda \Big( \mathbb{E}[T_L(X)] + c_\pi \Big),$$

*where $D(\cdot\|\cdot)$ is a statistical divergence, $T_S(X)$ and $T_L(X)$ denote the token lengths of the short and long CoT reasoning paths, $\lambda > 0$ is the per-token cost, and $c_\pi \geq 0$ is a fixed overhead for invoking*

*the long path. Then the optimal long-to-short adaptation policy can be:*

$$\pi^*(X) = \begin{cases} 0 & \text{if } J_S(X) < J_L(X) \quad (\text{choose short CoT}), \\ 1 & \text{otherwise} \quad (\text{choose long CoT}). \end{cases} \tag{3}$$

Theorem 2 establishes that an adapting strategy always exists between the short and long CoT paths, and is uniquely determined by a trade-off between the divergence of their predictive distributions and the token cost during inference and after training. Thus, the decision boundary is well-defined and deterministic almost everywhere in the adaptation policy. Specifically, as shown in Theorem 1, concatenating long and short CoT paths reduces the distributional divergence term, while the use of `<EASY>` tokens enables the model to approximate the optimal adaptation policy by balancing divergence reduction against token cost during inference. In particular, Theorem 2 also establishes that optimal adaptation depends on balancing the divergence between long and short predictive distributions with the per-token inference cost.

### 3.4 AUTOL2S-PLUS FINE-TUNING

Leveraging the ability of AutoL2S to dynamically adjust between short and long reasoning, we further extend to AutoL2S-Plus to enhance efficiency in LLM reasoning. AutoL2S-Plus incorporates a off-policy length-aware reinforcement learning objective that explicitly encourages shorter reasoning when appropriate. Motivated by O1-Pruner (Luo et al., 2025), we adopt the Length-Harmonizing Fine-Tuning objective, but employ different rewarding lengths guided by $f^*(\cdot|\theta_D)$. We formally define the length-aware loss with the reference model $f^*(\cdot|\theta_D)$ as:

$$L^{\text{Plus}}(\theta; x, y) = -\mathbb{E}_{x \sim \mathcal{D}, \, y \sim \pi_{\text{ref}}(y|x)} \Big[ \min \big( r(\theta) R_{\text{Plus}}(x, y|f^*), \, \text{clip}(r(\theta), 1-\epsilon, 1+\epsilon) R_{\text{Plus}}(x, y|f^*) \big) \Big],$$

where $r(\theta)$ denotes the likelihood ratio between the target policy $\theta$ and the reference policy $\pi_{\text{ref}}$, $R_{\text{Plus}}(x, y|f^*)$ is the reward to length of reasoning of $f^*(\cdot|\theta_D)$, and clip() is the clipping function.

Using a more accurate reference model to derive the expected reasoning length yields stronger compression and improves efficiency without degrading accuracy. We leverage the long- and short-form outputs of the fine-tuned AutoL2S model—used as the reference model—to estimate the expected average length and accuracy across the mixture of CoT paths. This objective encourages the model to harmonize long and short reasoning by rewarding generations that match the target length distribution while preserving correctness.

## 4 EXPERIMENTS

In this section, we conduct experiments to evaluate the performance of AutoL2S framework, aiming to answer the following three research questions: **RQ1**: How does AutoL2S perform on LLM reasoning tasks in terms of accuracy and efficiency? **RQ2**: Does the proposed long-short reasoning annotation contribute to effective length compression of the reasoning path during training? **RQ3**: What mechanisms enable auto long-short reasoning to preserve reasoning performance despite reduced output length?

### 4.1 DATASETS AND BASELINES

**Datasets** We train the AutoL2S framework on the Bespoke-Stratos-17k dataset (Labs, 2025) and evaluate it on six reasoning benchmarks: Math500 (Hendrycks et al., 2021), GPQA-Diamond (GPQA) (Rein et al., 2024), GSM8K (Cobbe et al., 2021), OlympiadBench-Math (He et al., 2024), AIME25 . Additional dataset statistics and preprocessing details are provided in Appendix D.

**Baseline Methods** We compare AutoL2S framework with the three state-of-the-art baselines to assess the effectiveness of length reduction and performance preservation. The baselines are listed as follows: (1) R1-Distilled reasoning LLMs (Bespoke-Stratos-3B/7B) (Yeo et al., 2025): LLMs fine-tuned in a supervised manner using the Bespoke-Stratos-17k reasoning dataset, which serves as an oracle for reasoning. (2) O1-pruner (Luo et al., 2025): introduces a Length-Harmonizing Reward, integrated with a PPO-style loss, to reduce the length of generated CoT reasoning. (3) CoT-Valve (Ma et al., 2025a): controls the length of reasoning by combining the LoRA weights

of distilled long-form reasoning CoT and non-reasoning model. (4) DPO (Rafailov et al., 2023): finetunes on the same aligned long-short CoT pairs. For each example, the short CoT is preferred if correct; otherwise, the long one is chosen, ensuring alignment with efficient and accurate reasoning goals (5) TokenSkip (Xia et al., 2025): trains on compressed CoT paths with mixed compression ratios. More details on baselines can be found in Appendix E.

## 4.2 Experimental Settings

In this section, we present the experimental settings used to train and assess AutoL2S. The following outlines the evaluation metrics and corresponding implementation details.

**Evaluation of Efficient LLM Reasoning.** Following the settings of (Luo et al., 2025; Yeo et al., 2025), we evaluate the efficiency of the reasoning task from two perspectives: (1) accuracy and (2) length of generated tokens. The ideal outcome is to maintain reasoning performance while minimizing the number of output tokens required for reasoning. Given the autoregressive decoding nature of LLMs, a shorter output CoT reasoning path directly leads to faster inference. Thus, in this work, we use the length of tokens as a metric to evaluate the efficiency of LLM reasoning.

**Implementation Details.** To demonstrate the flexibility of AutoL2S across different LLM backbones, we train the framework using two non-reasoning base LLMs: Llama3.2-3B-Instruct (Touvron et al., 2023) and Qwen2.5-7B-Instruct. The short reasoning samples are generated via rejection sampling with sampling numbers 4 and 8 using the math-capable Qwen2.5-Math-7B-Instruct model, with the inference temperature fixed at 0.7, following the settings of (Yeo et al., 2025; Yang et al., 2025). We filter out duplicate question–answer pairs that appear with both `<Easy>` and `<Long Trigger>` after rejection sampling, retaining only the pairs associated with `<Easy>` in such cases. For AutoL2S-plus training, we estimate the expected average reasoning length by sampling 16 generations per input under AutoL2S with a rejection sample size of 8. More details are in Appendix G.

## 4.3 Reasoning Efficiency of AutoL2S (RQ1)

We compare AutoL2S with the baseline methods in reasoning tasks. The results are presented in Table 1, showing the reasoning accuracy and output length for the models. Additional results from repetition experiments are provided in Appendix J and C. The purple cells represent the best performance, and blue cells refers to the second best among the settings. We calculate the improvement percentile relative to the Bespoke-Stratos-3B/7B model, a strong baseline finetuned on the Bespoke-Stratos-17k dataset. We conclude observations as follows:

- **Baseline Comparison.** AutoL2S outperforms CoT-Valve (Ma et al., 2025a) with better accuracy preservation and shorter reasoning path; and achieves shorter reasoning path than O1-pruner (Luo et al., 2025) with competitive accuracy preservation on average in four reasoning datasets. While both O1-pruner and the proposed AutoL2S are able to preserve reasoning accuracy, AutoL2S achieves approximately 4X shorter reasoning paths compared to O1-pruner. Furthermore, AutoL2S achieves nearly identical average reasoning accuracy compared to the oracle SFT R1-Distilled reasoning LLMs, while producing significantly shorter reasoning paths. This demonstrates that AutoL2S attains competitive performance in efficient reasoning tasks.

- **AutoL2S-Plus Comparison.** We compare AutoL2S-Plus with both the baseline and the base AutoL2S framework. AutoL2S-Plus achieves up to a 68.9% reduction in reasoning length without degrading accuracy, demonstrating the effectiveness of length-aware fine-tuning. Relative to AutoL2S, it further compresses the reasoning path while preserving task performance.

- **Rejection Sampling.** We find that moderate rejection sampling (e.g., rj = 4) achieves nearly 2X reduction in reasoning length with negligible accuracy loss compared to rj = 0. This reveals a sweet spot where the training distribution is dense enough to capture efficiency benefits without sacrificing correctness. At higher rj (e.g., 8), reasoning length is further compressed but at the cost of small accuracy drops. These results indicate that AutoL2S is not merely truncating reasoning steps; rather, it learns to generalize efficiency from curated supervision, producing compressed reasoning trajectories that respect correctness guarantees.

Table 1: Accuracy (Acc) and Token Length (Len) across five reasoning benchmarks for 7B models. "rj" indicates the rejection-sampling count for long-short annotation. AutoL2S-Plus (rj=0) uses AutoL2S (rj=0) as base model and reference model. Values in parentheses denote the accuracy improvement and token reduction relative to the Bespoke-Stratos-3B/7B model. Purple cells highlight the best value in each metric column, and blue cells highlight the second-best. For ties, ranking is resolved by prioritizing higher accuracy or lower token usage.

| | Average | | MATH500 | | GPQA | | GSM8K | | Olympiad | | AIME | |
|---|---|---|---|---|---|---|---|---|---|---|---|---|
| | Acc | Len | Acc | Len | Acc | Len | Acc | Len | Acc | Len | Acc | Len |
| *Llama-3.2-3B-Instruct* | | | | | | | | | | | | |
| Llama-3.2-3B-Instruct | 0.393 | 890 | 0.404 | 740 | 0.293 | 498 | 0.729 | 203 | 0.147 | 2117 | 0.000 | 887 |
| Bespoke-Stratos-3B | 0.479 | 9015 | 0.574 | 10148 | 0.273 | 8888 | 0.822 | 1387 | 0.246 | 15635 | 0.000 | 22677 |
| CoT-Valve | 0.422 (-0.057) | 10349 (+14.8%) | 0.478 (-0.096) | 10890 (+7.3%) | 0.283 (+0.010) | 9634 (+8.4%) | 0.773 (-0.049) | 2238 (+61.4%) | 0.154 (-0.092) | 18634 (+19.2%) | 0.000 (+0.000) | 24736 (+9.1%) |
| O1-pruner | 0.481 (+0.002) | 5043 (-44.1%) | 0.562 (-0.012) | 5295 (-47.8%) | 0.308 (+0.035) | 5394 (-39.3%) | 0.816 (-0.006) | 860 (-38.0%) | 0.236 (-0.010) | 8622 (-44.9%) | 0.033 (+0.033) | 14614 (-35.6%) |
| DPO | 0.479 (+0.000) | 5864 (-35.0%) | 0.574 (+0.000) | 5363 (-47.2%) | 0.283 (+0.010) | 6740 (-24.2%) | 0.832 (+0.010) | 911 (-34.3%) | 0.227 (-0.019) | 10441 (-33.2%) | 0.033 (+0.033) | 13856 (-38.9%) |
| TokenSkip | 0.441 (-0.038) | 9464 (+5.0%) | 0.512 (-0.062) | 10327 (+1.8%) | 0.258 (-0.015) | 9438 (+6.2%) | 0.801 (-0.021) | 2238 (+61.4%) | 0.191 (-0.055) | 15853 (+1.4%) | 0.000 (+0.000) | 21908 (-3.4%) |
| **AutoL2S (rj=0)** | **0.492 (+0.014)** | 6904 (-23.4%) | 0.552 (-0.022) | 5990 (-41.0%) | **0.389 (+0.116)** | 7520 (-15.4%) | 0.823 (+0.001) | 1166 (-15.9%) | 0.206 (-0.040) | 12941 (-17.2%) | 0.000 (+0.000) | 21248 (-6.3%) |
| **AutoL2S-Plus (rj=0)** | 0.467 (-0.012) | 3012 (-66.6%) | 0.534 (-0.040) | 3645 (-64.1%) | 0.273 (+0.000) | 2920 (-67.1%) | **0.829 (+0.007)** | 650 (-53.1%) | 0.230 (-0.016) | 4832 (-69.1%) | 0.000 (+0.000) | 6714 (-70.4%) |
| **AutoL2S (rj=4)** | 0.474 (-0.005) | 6680 (-25.9%) | **0.574 (+0.000)** | 5666 (-44.2%) | 0.283 (+0.010) | 7546 (-15.1%) | 0.812 (-0.010) | 1322 (-4.7%) | 0.226 (-0.021) | 12185 (-22.1%) | 0.033 (+0.033) | 18579 (-18.1%) |
| **AutoL2S-Plus (rj=4)** | 0.478 (-0.001) | 2220 (-75.4%) | 0.560 (-0.014) | 2090 (-79.4%) | 0.313 (+0.040) | 2175 (-75.5%) | 0.812 (-0.010) | 589 (-57.5%) | 0.226 (-0.020) | 4025 (-74.3%) | 0.033 (+0.033) | 4990 (-78.0%) |
| **AutoL2S (rj=8)** | **0.483 (+0.004)** | 5518 (-38.8%) | 0.546 (-0.028) | 4181 (-58.8%) | **0.369 (+0.096)** | 6165 (-30.6%) | 0.800 (-0.022) | 1021 (-26.4%) | 0.218 (-0.028) | 10706 (-31.5%) | 0.000 (+0.000) | 19422 (-14.4%) |
| **AutoL2S-Plus (rj=8)** | 0.467 (-0.012) | 1830 (-79.7%) | 0.550 (-0.024) | 1819 (-82.1%) | 0.273 (+0.000) | 2048 (-77.0%) | 0.810 (-0.012) | 353 (-74.5%) | 0.233 (-0.013) | 3099 (-80.2%) | 0.000 (+0.000) | 5072 (-77.9%) |
| *Qwen-2.5-7B-Instruct* | | | | | | | | | | | | |
| Qwen2.5-7B-Instruct | 0.495 | 728 | 0.748 | 556 | 0.308 | 27 | 0.902 | 260 | 0.384 | 896 | 0.133 | 1902 |
| Bespoke-Stratos-7B | 0.544 | 8139 | 0.824 | 5383 | 0.359 | 6049 | 0.926 | 1321 | 0.444 | 11322 | 0.167 | 16619 |
| CoT-Valve | 0.495 (-0.049) | 6306 (-22.5%) | 0.730 (-0.094) | 4483 (-16.7%) | 0.369 (+0.010) | 4930 (-18.5%) | 0.898 (-0.028) | 928 (-29.7%) | 0.378 (-0.066) | 8647 (-23.6%) | 0.100 (-0.067) | 12540 (-24.5%) |
| O1-pruner | 0.547 (+0.003) | 7797 (-4.2%) | **0.832 (+0.008)** | 5104 (-5.2%) | 0.399 (+0.040) | 5312 (-12.2%) | **0.936 (+0.010)** | 1065 (-19.4%) | 0.433 (-0.011) | 9586 (-15.3%) | 0.133 (-0.034) | 17920 (+7.8%) |
| DPO | 0.556 (+0.012) | 6060 (-25.5%) | 0.806 (-0.018) | 3688 (-31.5%) | 0.374 (+0.015) | 5961 (-1.5%) | 0.920 (-0.006) | 1576 (+19.3%) | 0.447 (+0.003) | 7364 (-35.0%) | **0.233 (+0.066)** | 11712 (-29.5%) |
| TokenSkip | 0.552 (+0.008) | 7944 (-2.4%) | **0.826 (+0.002)** | 5335 (-0.9%) | 0.434 (+0.075) | 5508 (-9.0%) | 0.918 (-0.008) | 1165 (-11.8%) | 0.447 (+0.003) | 10947 (-3.3%) | 0.133 (-0.034) | 16767 (+0.9%) |
| AlphaOne | 0.495 (-0.049) | 4856 (-40.3%) | 0.732 (-0.092) | 3867 (-28.2%) | 0.313 (-0.046) | 6278 (+3.8%) | 0.907 (-0.019) | 1943 (+47.1%) | 0.356 (-0.088) | 5252 (-53.6%) | 0.167 (+0.000) | 6940 (-58.2%) |
| **AutoL2S (rj=0)** | **0.561 (+0.017)** | 6886 (-15.4%) | 0.800 (-0.024) | 3468 (-35.6%) | **0.434 (+0.075)** | 4777 (-21.0%) | **0.934 (+0.008)** | 735 (-44.4%) | **0.470 (+0.026)** | 9068 (-19.9%) | 0.167 (+0.000) | 16384 (-1.4%) |
| **AutoL2S-Plus (rj=0)** | 0.534 (-0.010) | 2516 (-69.1%) | 0.782 (-0.042) | 1762 (-67.3%) | 0.419 (+0.060) | 2993 (-50.5%) | 0.923 (-0.003) | 718 (-45.6%) | 0.414 (-0.030) | 3240 (-71.4%) | 0.133 (-0.034) | 3865 (-76.7%) |
| **AutoL2S (rj=4)** | 0.550 (+0.006) | 5872 (-27.8%) | 0.786 (-0.038) | 2560 (-52.4%) | 0.409 (+0.050) | 3495 (-42.2%) | 0.917 (-0.008) | 509 (-61.5%) | 0.438 (-0.006) | 7991 (-29.4%) | 0.200 (+0.033) | 14807 (-10.9%) |
| **AutoL2S-Plus (rj=4)** | 0.556 (+0.012) | 2170 (-73.3%) | 0.798 (-0.026) | 1627 (-69.8%) | 0.409 (+0.050) | 2401 (-60.3%) | 0.926 (+0.000) | 661 (-50.0%) | **0.447 (+0.003)** | 3023 (-73.3%) | 0.200 (+0.033) | 3137 (-81.1%) |
| **AutoL2S (rj=8)** | 0.538 (-0.006) | 5141 (-36.8%) | 0.798 (-0.026) | 2146 (-55.1%) | 0.394 (+0.035) | 3492 (-42.3%) | 0.929 (+0.003) | 488 (-63.1%) | 0.436 (-0.008) | 6459 (-43.0%) | 0.133 (-0.033) | 12852 (-22.7%) |
| **AutoL2S-Plus (rj=8)** | **0.558 (+0.014)** | 2531 (-68.9%) | 0.820 (-0.004) | 1719 (-68.1%) | 0.424 (+0.065) | 3485 (-42.4%) | 0.920 (-0.006) | 880 (-33.4%) | 0.424 (-0.020) | 3041 (-73.1%) | 0.200 (+0.033) | 3528 (-78.8%) |

## 4.4 IMPACT ON LONG-SHORT REASONING ANNOTATION (RQ2)

In this section, we analyze the impact of our concatenation strategy to combine long and short CoT reasoning paths in the long-short reasoning adaptation process. We conduct ablation studies on different distillation strategies for long-short CoT reasoning paths, with Qwen2.5-7B-Instruct model serving as the non-reasoning base model. We compare three other different format of annotation to the proposed Long-to-short Reasoning Annotation (i.e., *Long-short Distill*): (1) *Long-only Distill* represents the original distillation from only long reasoning in Bespoke-Stratos-17k reasoning dataset, following the format in Equation 5; (2) *Short-long Distill* switches the position of long and short reasoning path in Equation 4; and (3) *Long-short Separated Distill* constructs the long and short CoT reasoning paths following the format in Equation 5, where short CoT reasoning paths are replaced with long reasoning paths only whenever the corresponding answers are correct. All results are demonstrated in Table 3. Compared with other formats of long-short term annotation, we observe that *Long-Short Distill* achieves the best performance in terms of accuracy preservation and output length.

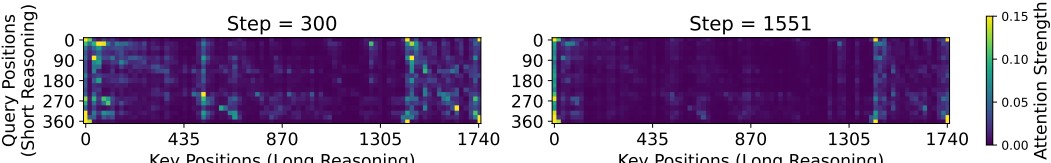

Figure 2: Comparison of attention maps at early and late training steps of AutoL2S. Step 1551 corresponds to the final training step. Given the long sequence lengths, we group every 20 tokens together to calculate attention scores between long and short reasoning paths for better visualization.

### 4.5 IMPACT OF THE <EASY> TOKEN (RQ2)

In this section, we examine the impact of the <EASY> token on enhancing both the efficiency and efficacy of LLM reasoning tasks. The results are showcased in Table 4. We conduct the ablation studies on three different cases in terms of the long-short triggers and <EASY> tokens that we utilize in the AutoL2S framework. Based on the AutoL2S framework, (1) "w/ Force-Short" refers to the setting where <Short Trigger> is always used to initiate reasoning path generation; (2) "w/ Force-Long" denotes the setting where <Longer Trigger> is consistently used to initiate CoT generation; and (3) "w/o <EASY>" indicates that no explicit trigger is applied and the model generates reasoning paths in formats that follow either Equation 5 or Equation 4. We summarize the findings as follows:

- **AutoL2S vs. "w/o <EASY>":** AutoL2S outperforms the "w/o <EASY>" variant in both reasoning accuracy and in the length of the generated CoT reasoning paths. This further demonstrates that incorporating the <EASY> token to automatically switch between easy and regular reasoning modes improves efficiency without compromising performance.
- **AutoL2S vs. "Force-Long":** Compared to the "Force-Long" case, AutoL2S obtains a similar reasoning accuracy on average while generating around 30% shorter of the reasoning length. Furthermore, compared to "Force-Long" with Bespoke-Stratos-7B, trained on the entire long CoT reasoning data, we can observe that "Force-Long" outperforms Bespoke-Stratos-7B in terms of reasoning accuracy while holding similar reasoning path length. These results indicate that the long reasoning paths generated by our method are of higher quality than those produced by Bespoke-Stratos-7B.

### 4.6 MECHANISM BEHIND THE AUTO LONG-SHORT REASONING (RQ3)

In this section, we discuss the mechanism explanation of AutoL2S training. To assess the mechanism behind, Figure 2 presents the attention map comparisons across different training steps of AutoL2S, highlighting the benefit of the concatenation order used in *Long-Short Distill*. In the early stages of training (i.e., Figure 2 left side: training step 300), we observe that long CoT reasoning paths significantly impact the attention patterns of short CoT reasoning paths, indicating that long-form reasoning benefits the learning of short reasoning generation. As training progresses till the end (i.e., Figure 2 right side: training step 1551), the correlation between long and short CoT reasoning paths significantly diminishes, indicating that they evolve into two distinct components. This separation further explains why Auto Long-Short Reasoning is effective and flexible in switching to easy questions simply using the <Short Trigger> when the <EASY> token is presented during inference. The phenomenon again meets the properties of Theorem 1, where long CoT reasoning paths provide auxiliary information for short-path learning. This also explain the reason why the direct use of <Short Trigger> remains effective, without introducing dummy key–value pairs or modifying positional encodings.

## 5 CONCLUSION

This paper presents the Auto Long-Short Reasoning (AutoL2S), a dynamic and model-agnostic framework for improving the efficiency of LLM reasoning. By training on proposed annotated data

that pairs long and short CoT reasoning paths and incorporating a special `<EASY>` token, AutoL2S enables LLMs to decide when extended reasoning is necessary and when a concise path suffices. This learned adaptive behavior helps avoid overthinking simple questions, reducing unnecessary computation. Experimental results show that AutoL2S reduces reasoning length by up to 70% without degrading performance, demonstrating its effectiveness for scalable and cost-efficient LLM deployment in real-world settings.

## ETHICS STATEMENT

This research does not involve human subjects, personally identifiable data, or sensitive attributes. All datasets used are publicly available benchmark datasets (e.g., math word problems, programming tasks) that have been widely adopted in prior work. We are not aware of any potential harms associated with releasing our methods, as with any AI technology, misuse could occur if applied irresponsibly in high-stakes settings. We encourage careful consideration of societal impacts and safe deployment practices.

## REPRODUCIBILITY STATEMENT

We follow ICLR's reproducibility guidelines. All datasets used in this study are publicly available, and we provide detailed dataset descriptions. The proposed method is implemented using standard libraries, and we have provided the codebase, including training scripts, hyperparameter settings, and evaluation procedures for paper review purposes.

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

## A    USE OF LLMS

We used a large language model only for spelling and grammar correction of the manuscript text. The LLM was not involved in research ideation, experimental design, data generation, analysis, or substantive writing beyond copy-editing. All content and claims were authored and verified by the authors, who take full responsibility for the paper. The LLM is not an author.

## B    DETAILS OF AUTOL2S ANNOTATION

We provide the complete data formats used to annotate both long and short CoT reasoning paths. These formats serve as templates for generating training samples in AutoL2S.

**Short CoT Reasoning Paths for EASY Questions.**    EASY questions include both long and short reasoning paths. The `<EASY>` token indicates that the question is solvable through a short reasoning path. `<Long Trigger>` and `<Short Trigger>` mark the start of the long and short reasoning, and `<Answer Trigger>` marks the start of the answer. The complete data format for the EASY questions is given as follows:

$$
\text{Question} \ \texttt{<EASY>} \ \texttt{<Long Trigger>} \ \text{Long reasoning path} \ \texttt{<Answer>} \ \text{Final answer}
$$
$$
\texttt{<Short Trigger>} \ \text{Short reasoning path} \ \texttt{<Answer Trigger>} \ \text{Final answer} . \tag{4}
$$

**Long CoT Reasoning Paths.**    Non-EASY questions are annotated only with the long reasoning path. Similarly, `<Long Trigger>` marks the beginning of the long reasoning and `<Answer Trigger>` marks the start of the answer. The format is:

$$
\text{Question} \ \texttt{<Long Trigger>} \ \text{Long reasoning path} \ \texttt{<Answer Trigger>} \ \text{Final answer} . \tag{5}
$$

These annotations provide both complete and concise reasoning trajectories and allow for the distillation of the decision-making capabilities of DeepSeek-R1 into the target model.

## C    ADDITIONAL RESULTS OF AUTOL2S ON QWEN2.5-3B-INSTRUCT

We here provide the additional results of AutoL2S on Qwen2.5-3B-Instruct under five datasets. As shown in Table 2, AutoL2S obtains 42% shorter reasoning paths without sacrificing performance.

Table 2: Accuracy (Acc) and Token Length (Len) across five reasoning benchmarks for models based on Qwen2.5-3B-Instruct models.

| | Average | | MATH500 | | GPQA | | GSM8K | | Olympiad | | AIME | |
|---|---|---|---|---|---|---|---|---|---|---|---|---|
| | Acc | Len | Acc | Len | Acc | Len | Acc | Len | Acc | Len | Acc | Len |
| *Qwen2.5-3B-Instruct* | | | | | | | | | | | | |
| Qwen2.5-3B-Instruct | 0.362 | 766 | 0.622 | 806 | 0.349 | 770 | 0.679 | 376 | 0.266 | 1158 | 0.100 | 872 |
| Bespoke-Stratos-3B | 0.383 | 12739 | 0.636 | 9246 | 0.308 | 10129 | 0.848 | 1624 | 0.272 | 14724 | 0.100 | 20217 |
| CoT-Valve | | | 0.602 | 4980 | 0.258 | 6898 | 0.805 | 1660 | 0.270 | 10017 | 0.000 | 11059 |
| | (-22.5%) | (-33.2%) | (-5.3%) | (-46.1%) | (-16.4%) | (-31.9%) | (-5.1%) | (+2.2%) | (-0.6%) | (-32.0%) | (-100.0%) | (-45.3%) |
| O1-pruner | | | 0.704 | 6769 | 0.283 | 7348 | 0.859 | 1210 | 0.295 | 11416 | 0.100 | 15022 |
| | (+8.5%) | (-29.0%) | (+10.7%) | (-26.8%) | (-8.1%) | (-27.5%) | (+1.2%) | (-25.5%) | (+8.8%) | (-22.5%) | (0.0%) | (-25.7%) |
| DPO | | | 0.684 | 5238 | 0.323 | 7533 | 0.824 | 1529 | 0.298 | 10952 | 0.067 | 14005 |
| | (-8.7%) | (-29.5%) | (+7.5%) | (-43.4%) | (+4.8%) | (-25.6%) | (-2.9%) | (-5.8%) | (+9.8%) | (-25.6%) | (-33.0%) | (-30.7%) |
| TokenSkip | | | 0.526 | 9100 | 0.263 | 16083 | 0.805 | 2237 | 0.178 | 16881 | 0.033 | 26126 |
| | (-25.5%) | (+15.5%) | (-17.3%) | (-1.6%) | (-14.6%) | (+58.8%) | (-5.1%) | (+37.8%) | (-34.4%) | (+14.6%) | (-67.0%) | (+29.2%) |
| Ours (rj=0) | | | 0.694 | 5976 | 0.253 | 6252 | 0.839 | 1346 | 0.280 | 13314 | 0.100 | 19166 |
| | (+0.9%) | (-25.3%) | (+9.1%) | (-35.4%) | (-18.0%) | (-38.3%) | (-1.2%) | (-17.1%) | (+3.3%) | (-9.6%) | (0.0%) | (-5.2%) |
| Ours (rj=4) | | | 0.638 | 4202 | 0.298 | 5357 | 0.840 | 899 | 0.270 | 10410 | 0.033 | 18473 |
| | (-10.7%) | (-38.3%) | (+0.3%) | (-54.6%) | (-3.3%) | (-47.1%) | (-1.0%) | (-44.6%) | (-0.6%) | (-29.3%) | (-67.0%) | (-8.6%) |
| Ours (rj=8) | | | 0.636 | 3928 | 0.253 | 5431 | 0.826 | 741 | 0.273 | 9460 | 0.100 | 17797 |
| | (+0.5%) | (-42.0%) | (0.0%) | (-57.5%) | (-18.0%) | (-46.4%) | (-2.7%) | (-35.8%) | (+0.6%) | (-35.8%) | (0.0%) | (-12.0%) |

# D    DETAILS OF DATASET

We train the AutoL2S framework under Bespoke-Stratos-17k (Labs, 2025) dataset and assess the framework on long-to-short reasoning task under four different reasoning datasets. The details of the assessment datasets are provided as follows:

- **Math500** (Hendrycks et al., 2021): A challenging benchmark consisting of 500 high-quality math word problems that require multi-step symbolic reasoning.
- **GPQA-Diamond (GPQA)** (Rein et al., 2024): The Graduate-Level Physics Question Answering (GPQA) dataset contains 198 multiple-choice questions from graduate-level physics exams.
- **GSM8K** (Cobbe et al., 2021): A widely-used benchmark comprising 1319 grade school-level math word problems.
- **Olympiad Bench Math (Olympiad)** (He et al., 2024): A collection of 674 math competition problems inspired by middle and high school mathematics Olympiad competitions.
- **AIME25**: A benchmark based on problems from the American Invitational Mathematics Examination, comprising 25 challenging questions that require concise yet deep reasoning steps.
- **LiveCodeBench V2**: A programming-oriented benchmark consisting of live coding tasks that assess reasoning ability in code synthesis, debugging, and execution.

# E    DETAILS OF BASELINE IMPLEMENTATION

## E.1    BESPOKE-STRATOS

We implement this baseline by fully fine-tuning language models on the Bespoke-Stratos-17k dataset, which comprises 17,000 examples of questions, long-form reasoning traces, and corresponding answers. The resulting model serves as an oracle reference for reasoning performance.

Following standard SFT procedures, training is performed by minimizing the standard cross-entropy loss over the input sequence. We employ the AdamW optimizer with a learning rate of $1e-5$ and a batch size of 32. Fine-tuning is conducted for three epochs on two NVIDIA A100 80GB GPUs with mixed-precision training enabled. For the 7B base model, we directly utilize the publicly released checkpoint VanWang/Bespoke-Stratos-7B-repro-SFT.

## E.2    O1-PRUNER

O1-pruner introduces a Length-Harmonizing Reward, integrated with a PPO-style loss, to optimize the policy model $\pi_\theta$ and reduce the length of generated chain-of-thought (CoT) reasoning. Considering the effectiveness of off-policy training with pre-collected data, O1-pruner adopts an off-policy training approach by sampling from the reference model $\pi_{\text{ref}}$ rather than from $\pi_\theta$. Specifically, the training procedure consists of two steps: (1) generating CoT samples using $\pi_{\text{ref}}$, and (2) fine-tuning the policy model with the proposed PPO-style objective based on the generated samples.

In our implementation, we follow the original experimental setting and reproduce the method based on its official repository.[1] For training, we sample 5,000 problems from the Bespoke-Stratos-17k dataset and generate 16 solutions for each problem. We then perform length-harmonizing fine-tuning for one epoch to jointly optimize both output length and answer correctness. To ensure fair comparison with our method, we use Bespoke-Stratos-3B/7B as the reference model and set the maximum sequence length to 10,240 tokens when training.

## E.3    COT-VALVE

COT-Valve is designed to enable models to generate reasoning chains of varying lengths. It controls the length of reasoning by linearly combining the LoRA weights of distilled long-form reasoning CoT and non-reasoning model. For the specifc Long to Short CoT task, it has three stages: (1) finetune the LLM base model on a long-cot dataset using Lora to identify a direction in the parameter

---

[1]`https://github.com/StarDewXXX/O1-Pruner`

space that control the length of generated CoT(2) merge Lora weights with the base model at varying interpolation ratios generate models and use them construct datasets containing CoT of decreasing lengths (3) finetuning the distilled reasoning model with the generated dataset in a progressive way, where the model is trained with shorter reasoning path samples between epochs. This progressive training strategy enables the model to gradually compress its reasoning while maintaining correctness.

In our implementation, we follow the original configuration in CoT-Valve. The LoRA rank and LoRA alpha are set to 32 and 64, respectively, for both the first and third stages. In the first stage, we finetune the non-reasoning models Qwen2.5-3B-Instruct/Qwen2.5-7B-Instruct on the Bespoke-Stratos-17k dataset for three epochs using Lora. The learning rate is 4e-5 and the batch size is 64. In the second stage, we apply LoRA weight interpolation with coefficients 0.8 and 0.6. Due to resource constraints, we randomly sample 2,000 questions for each interpolated model to generate responses, and retain only those samples with correct answers. In the third stage, the model that we get in the first stage is further fine-tuned for 2 epochs on each type of generated dataset, using the same learning rate of 4e-5 and a batch size of 64.

# F  PROOF OF THEOREM

In this section, we present and prove Theorem 1 and Theorem 2, with accompanying remarks to provide intuitive explanations.

**Theorem 1 (Concatenation Advantage for Long–Short CoT Training).** *Let $X$ denote the input, $L = (\ell_1, \ldots, \ell_{T_L})$ the long-CoT token sequence, and $S = (s_1, \ldots, s_{T_S})$ the short-CoT token sequence, with training order $L$ to $S$. Then, the conditional entropy $H(\cdot | \cdot)$ of the next short token satisfies:*

$$H(S_t \mid X, L, S_{<t}) \ \leq \ H(S_t \mid X, S_{<t}), \quad \forall\, t \in [1, T_S]. \tag{6}$$

*Equivalently, averaging across all positions with the improvement quantified as*

$$\frac{1}{T_S} \sum_{t=1}^{T_S} \Big[ H(S_t \mid X, S_{<t}) - H(S_t \mid X, L, S_{<t}) \Big] = \frac{1}{T_S} \sum_{t=1}^{T_S} I(S_t; L \mid X, S_{<t}) \ \geq 0. \tag{7}$$

*Thus, the long CoT path $L$ provides additional mutual information $I(\cdot | \cdot)$ that strictly increases the entropy of the short CoT path $S$ whenever $L$ is informative about $S$.*

*Proof.* The inequality follows directly from the fact that conditioning reduces entropy: adding $L$ to the conditioning set cannot increase the uncertainty of $S_t$. Formally, for each $t \in [T_S]$,

$$H(S_t \mid X, L, S_{<t}) \ \leq \ H(S_t \mid X, S_{<t}).$$

Averaging over $t$ yields the stated inequality.

The gap between the two sides can be expressed as the conditional mutual information:

$$\frac{1}{T_S} \sum_{t=1}^{T_S} \big[ H(S_t \mid X, S_{<t}) - H(S_t \mid X, L, S_{<t}) \big] = \frac{1}{T_S} \sum_{t=1}^{T_S} I(S_t; L \mid X, S_{<t}) \ \geq 0.$$

In the realizable training case under long CoT path distillation, the model is optimized with the per-token cross-entropy objective

$$\mathrm{CE}(S \mid \mathcal{C}) = \frac{1}{T_S} \sum_{t=1}^{T_S} \mathbb{E}\big[ - \log p_\theta(s_t \mid \mathcal{C}, S_{<t}) \big],$$

where the context $\mathcal{C}$ is either $(X)$ or $(X, L)$. When $p_\theta$ matches the true distribution, the cross-entropy coincides with the entropy above. Thus, the same inequality carries over to cross-entropy:

$$\mathrm{CE}(S \mid X, L) \ \leq \ \mathrm{CE}(S \mid X),$$

with the gap equal to the average conditional mutual information. Finally, since perplexity is defined as $\mathrm{PPL}(S \mid \mathcal{C}) = \exp(\mathrm{CE}(S \mid \mathcal{C}))$, the inequality extends directly to perplexity:

$$\mathrm{PPL}(S \mid X, L) \ \leq \ \mathrm{PPL}(S \mid X).$$

□

**Theorem 2** (**Optimal Adaptation with `<EASY>` Token**). *Let $p_\theta^L(\cdot \mid x)$ and $p_\theta^S(\cdot \mid x)$ denote the predictive distributions when decoding with the long and short CoT paths $L = (\ell_1, \ldots, \ell_{T_L})$ and $S = (s_1, \ldots, s_{T_S})$, respectively. Given an input $x \in \mathcal{Y}$, define the per-instance risks as*

$$J_S(x) = \mathbb{E}\Big[D\Big(p_\theta^S(\cdot \mid x) \,\Big\|\, p_\theta^L(\cdot \mid x)\Big)\Big] \;+\; \lambda\,\mathbb{E}[T_S(x)], \tag{8}$$

$$J_L(x) = \lambda\Big(\mathbb{E}[T_L(x)] + c_\pi\Big), \tag{9}$$

*where $D(\cdot\|\cdot)$ is a statistical divergence, $T_S(x)$ and $T_L(x)$ denote the token lengths of the short and long CoT paths, $\lambda > 0$ is the per-token cost, and $c_\pi \geq 0$ is a fixed overhead for invoking the long path. Then the optimal adaptation policy is*

$$\pi^*(x) = \begin{cases} 0 & \text{if } J_S(x) < J_L(x) \quad (\text{choose short}), \\ 1 & \text{otherwise} \quad (\text{choose long}). \end{cases} \tag{10}$$

*Proof.* We provide the proof within the following six steps.

**Assumptions from AutoL2S Design.** Let $\mathcal{Y}$ be the input space with data distribution $\mathcal{D}$. Assume $D(\cdot\|\cdot) \geq 0$ is a statistical divergence for which $\mathbb{E}[D(p_\theta^S(\cdot \mid x) \| p_\theta^L(\cdot \mid x))]$ exists, and the token lengths $T_S(x), T_L(x)$ are nonnegative random variables with finite expectations. Let an adaptation policy be a measurable mapping $\pi : \mathcal{Y} \to \{0, 1\}$, where $\pi(x){=}0$ chooses short reasoning CoT and $\pi(x){=}1$ chooses long reasoning CoT. For a policy $\pi$, define the population risk

$$\mathcal{R}(\pi) := \mathbb{E}_{x \sim \mathcal{D}}\Big[ J_S(x)\,\mathbf{1}\{\pi(x) = 0\} + J_L(x)\,\mathbf{1}\{\pi(x) = 1\} \Big].$$

By the assumptions above, $\mathcal{R}(\pi)$ is well-defined and finite.

**Step 1 (Reduction to deterministic policies).** Consider any *randomized* policy that, for a fixed $x$, chooses short with probability $\alpha(x) \in [0, 1]$ and long with probability $1 - \alpha(x)$. Its *conditional* (on $x$) contribution to risk equals

$$\alpha(x)\,J_S(x) + (1 - \alpha(x))\,J_L(x) = J_L(x) + \alpha(x)\,\Delta(x), \quad \text{where } \Delta(x) := J_S(x) - J_L(x).$$

Since this expression is linear in $\alpha(x)$, its minimum over $\alpha(x) \in [0, 1]$ is always achieved at an extreme point $\alpha(x) \in \{0, 1\}$:

$$\alpha^\star(x) = \begin{cases} 1, & \text{if } \Delta(x) < 0, \\ 0, & \text{if } \Delta(x) > 0, \\ \text{any in } [0, 1], & \text{if } \Delta(x) = 0. \end{cases}$$

Hence randomization cannot improve over a deterministic rule, and it suffices and prove to optimize over deterministic policy $\pi$.

**Step 2 (Pointwise decomposition).** For any deterministic $\pi$,

$$\mathcal{R}(\pi) = \mathbb{E}\big[J_L(x)\big] + \mathbb{E}\big[\Delta(x)\,\mathbf{1}\{\pi(x) = 0\}\big].$$

The first term does not depend on $\pi$, so minimizing $\mathcal{R}(\pi)$ reduces to minimizing the second term. Because the expectation is taken with respect to $\mathcal{D}$ and the integrand depends on $\pi$ only through the indicator, this is a pointwise decision:

**Step 3 (Pointwise optimal action).** For a fixed $x$:

$$\min_{a \in \{0,1\}} \big\{ \Delta(x)\,\mathbf{1}\{a = 0\} \big\} = \begin{cases} \Delta(x), & \text{if } a = 0 \text{ and } \Delta(x) < 0, \\ 0, & \text{if } a = 1 \text{ or } \Delta(x) \geq 0, \end{cases}$$

which is achieved by choosing $a{=}0$ (short) when $\Delta(x) < 0$, and $a{=}1$ (long) otherwise. Thus the Bayes-optimal policy is

$$\pi^\star(x) = \begin{cases} 0, & \text{if } \Delta(x) < 0 \ \ \big(\text{i.e., } J_S(x) < J_L(x)\big), \\ 1, & \text{otherwise.} \end{cases}$$

This is exactly the threshold rule stated in the theorem.

**Step 4 (Existence and uniqueness).** Existence follows because the pointwise minimum is always attained by an action in $\{0, 1\}$. Uniqueness holds everywhere except on the *tie set* $\{x : \Delta(x) = 0\}$ where both actions yield the same risk; changing $\pi^\star$ on this set does not alter $\mathcal{R}(\pi^\star)$. Hence the optimal policy is unique almost surely (up to ties).

**Step 5 (Explicit threshold and interpretation).** Expanding $\Delta(x)$ gives

$$\Delta(x) = \underbrace{\mathbb{E}\Big[D\Big(p_\theta^S(\cdot \mid x) \,\Big\|\, p_\theta^L(\cdot \mid x)\Big)\Big]}_{\text{predictive distribution divergence}} + \lambda\Big(\mathbb{E}[T_S(x)] - \mathbb{E}[T_L(x)] - c_\pi\Big).$$

Thus $\pi^\star(x)=0$ (choose short) iff the divergence penalty is outweighed by the token savings:

$$\mathbb{E}\Big[D\Big(p_\theta^S \,\Big\|\, p_\theta^L\Big)\Big] \;<\; \lambda\Big(\mathbb{E}[T_L(x)] + c_\pi - \mathbb{E}[T_S(x)]\Big).$$

Equivalently, *choose short when predicted distributions are sufficiently close* and *the token savings are large enough*.

**Step 6 (Comparative statics).** The decision boundary moves monotonically: increasing $c_\pi$ or the long/short length gap $\mathbb{E}[T_L] - \mathbb{E}[T_S]$ makes short more favorable; increasing the divergence or decreasing the length gap makes long more favorable. Increasing $\lambda$ amplifies the weight on token savings, thus favoring short when $\mathbb{E}[T_L] + c_\pi > \mathbb{E}[T_S]$. $\qquad\square$

**Remark 1.** *Theorem 2 establishes that an optimal adaptation strategy between long and short CoT paths always exists and is essentially unique, reducing to a deterministic threshold rule. The policy selects the short path whenever the predictive distribution of the short rationale is sufficiently close to that of the long reasoning while offering enough token savings to offset the overhead of using the long path. This shows that the* <EASY> *token is not an ad hoc mechanism, but corresponds to a Bayes-optimal decision that balances semantic fidelity and inference efficiency. Together with Theorem 1, this highlights that the long reasoning paths not only improves the learnability of the short reasoning paths during training, but also guides optimal switching at inference time.*

## G   DETAILS OF IMPLEMENTATION AND INSTRUCTION PROMPT AND TRIGGERS

In this section, we introduce the format of instruction prompts and triggers that we utilized in our AutoL2S framework.

### G.1   DETAILS OF IMPLEMENTATION SETTINGS

All experiments for the 7B base model are conducted using four NVIDIA A100 80G GPUs, while those for the 3B base model utilize two NVIDIA A100 80GB GPUs. We leverage the Transformers library for fine-tuning and vLLM for efficient inference. Fine-tuning is performed using the AdamW optimizer with a learning rate of 1e-5. The temperature is fixed at 0.7 in both AutoL2S and baselines, ensuring that the output reasoning sequences are fully generated without truncation.

### G.2   SYSTEM PROMPT AND INFERENCE PROMPT

---

**System Prompt / Inference Prompt**

Your role as an assistant is to solve problems using one of two ways.
First, think step-by-step and give the answer. The solution should be formatted as follows:
```
<|begin_of_solution|>
{precise and concise solution}
<|end_of_solution|>
```

Second, your output thoroughly explores questions through a systematic long thinking process before providing the final precise and accurate solutions. This requires engaging in a comprehensive cycle of analysis, summarizing, exploration, reassessment, reflection, backtracing, and iteration to develop well-considered thinking process.

Please structure your response into two main sections: Thought and Solution.

In the Thought section, detail your reasoning process using the specified format:
```
<|begin_of_thought|>  {thought with steps separated with '\n\n'}
<|end_of_thought|>
```

Each step should include detailed considerations such as analysing questions, summarizing relevant findings, brainstorming new ideas, verifying the accuracy of the current steps, refining any errors, and revisiting previous steps.

In the Solution section, based on various attempts, explorations, and reflections from the Thought section, systematically present the final solution that you deem correct. The solution should remain a logical, accurate, concise expression style and detail necessary steps needed to reach the conclusion, formatted as follows:
```
<|begin_of_solution|>
{final formatted, precise, and clear solution}
<|end_of_solution|>
```

Now, try to solve the following question through the above guidelines:

---

### G.3   SPECIAL TOKEN AND LONG-SHORT TRIGGER

---

**Special Easy Token and Long-short Trigger**

`<Easy>` Token: `<specialLong>`

`Short Trigger`: This is a trigger to ensure the model's upcoming output `<short>`.

**Long Trigger**: Let's consider this problem in a `<pureLong>` way.

---

## H TABLE OF RQ2 AND RQ3

We showcase the tables of the impact of the `<EASY>` token and data annotation.

Table 3: Ablation studies of different annotation strategies on training the AutoL2S framework.

| Method | Average | | MATH500 | | GPQA | | GSM8K | | Olympiad | |
|---|---|---|---|---|---|---|---|---|---|---|
| | Acc | Len | Acc | Len | Acc | Len | Acc | Len | Acc | Len |
| Long-only Distill | 0.638 | 6019 | 0.824 | 5383 | 0.359 | 6049 | 0.926 | 1321 | 0.444 | 11322 |
| Long-short Separated Distill | 0.626 | 4250 | 0.798 | 3123 | 0.389 | 5096 | 0.919 | 864 | 0.399 | 7919 |
| Short-Long Distill | 0.610 | 1798 | 0.758 | 3239 | 0.414 | 1754 | 0.883 | 701 | 0.384 | 1499 |
| Long-Short Distill (rj = 0) | 0.660 | 4512 | 0.800 | 3468 | 0.434 | 4777 | 0.934 | 735 | 0.470 | 9068 |
| Long-Short Distill (rj = 8) | 0.639 | 3214 | 0.798 | 2416 | 0.394 | 3492 | 0.929 | 488 | 0.436 | 6459 |

Table 4: Ablation studies of auto long-short reasoning using `<EASY>` token.

| Method | Average | | MATH500 | | GPQA | | GSM8K | | Olympiad | |
|---|---|---|---|---|---|---|---|---|---|---|
| | Acc | Len | Acc | Len | Acc | Len | Acc | Len | Acc | Len |
| Qwen2.5-Math-7B-Instruct | 0.603 | 1072 | 0.792 | 798 | 0.288 | 1806 | 0.943 | 357 | 0.389 | 1328 |
| Bespoke-Stratos-7B | 0.638 | 6019 | 0.824 | 5383 | 0.359 | 6049 | 0.926 | 1321 | 0.444 | 11322 |
| AutoL2S w/o `<EASY>` | 0.644 | 6327 | 0.792 | 5999 | 0.399 | 6489 | 0.923 | 1389 | 0.463 | 11432 |
| AutoL2S w/ Force-Short | 0.639 | 1668 | 0.776 | 1616 | 0.409 | 1943 | 0.925 | 343 | 0.444 | 2768 |
| AutoL2S w/ Force-Long | 0.664 | 5912 | 0.844 | 5437 | 0.409 | 5808 | 0.922 | 1230 | 0.481 | 11173 |
| AutoL2S (rj = 0) | 0.660 | 4512 | 0.800 | 3468 | 0.434 | 4777 | 0.934 | 735 | 0.470 | 9068 |

# I RELATED WORK

**Reasoning-capable LLMs.** Recent advancements in LLMs have significantly enhanced their reasoning capabilities, exemplified by large reasoning models such as OpenAI o1 (OpenAI) and DeepSeek-R1 (Guo et al., 2025), and QwQ-32B (Team). OpenAI o1 (OpenAI) introduces advanced reasoning mechanisms designed to tackle complex problems, such as mathematical and programming tasks. Similarly, DeepSeek-R1 (Guo et al., 2025) enhances reasoning abilities by employing RL to incentivize effective reasoning behaviors. Additionally, DeepSeek-R1 curates specialized reasoning datasets, enabling the explicit distillation of reasoning capabilities into smaller models through SFT.

**Efficient LLM Reasoning.** Thinking steps of LLMs have become longer, leading to the "over-thinking problem" (Chen et al., 2024; Sui et al., 2025). To mitigate lengthy responses and reasoning processes, several works have been conducted to shorten the thinking steps and produce more concise reasoning (Sui et al., 2025). *RL-based methods* aim to encourage full-length reasoning models to generate concise thinking steps or train non-reasoning models to learn efficient reasoning by incorporating a length-aware reward (Team et al., 2025; Luo et al., 2025; Aggarwal & Welleck, 2025; Yeo et al., 2025; Shen et al., 2025; Hou et al., 2025). Specifically, they propose designing a length-based score to penalize excessively lengthy responses, complementing original rewards (e.g., format reward and accuracy reward). Kimi K1.5 (Team et al., 2025) calculates a length reward based on the response length relative to the shortest and longest responses. L1 (Aggarwal & Welleck, 2025) modifies the training data with the designated length constraint instruction, and then add the length reward. O1-Pruner (Luo et al., 2025) introduces the length-harmonizing reward, which calculates the ratio of lengths between the reference model and predicted model along with the accuracy-based constraints.

*SFT-based methods* curate variable-length CoT training datasets to fine-tune overthinking reasoning models for shorter reasoning paths or to equip non-reasoning models with efficient reasoning capabilities (Han et al., 2024; Xia et al., 2025; Ma et al., 2025a; Yu et al., 2025; Cui et al., 2025). Specifically, based on long CoT reasoning data, they curate shorter yet accurate CoT reasoning paths as training data. Token-skip (Xia et al., 2025) leverages LLMLingua (Jiang et al., 2023) to compress lengthy CoT responses into shorter ones based on semantic scores, and then fine-tunes the model for efficient reasoning. CoT-Valve (Ma et al., 2025a) controls the magnitude of LoRA (Hu et al., 2022) weights to generate variable-length CoT training data, which are then used to fine-tune an efficient reasoning model. Token-Budget (Han et al., 2024) assigns specific token budgets to prompts in order to generate shorter reasoning steps, and these concise CoT examples are then used for model fine-tuning.

## J    ROBUSTNESS ANALYTICS OF AUTOL2S

To assess the robustness of our method, we further evaluated AutoL2S on both 3B and 7B models under three different runs with different random seeds. The reported values correspond to the mean and standard deviation with same settings presented in Section 4. The **bold** numbers represent the best performance, and underline refers to the second best among the settings.

Based on the average performance, AutoL2S outperforms CoT-Valve by achieving higher accuracy and generating shorter reasoning paths. Compared to O1-pruner, AutoL2S produces shorter reasoning paths while maintaining comparable average accuracy across all four reasoning benchmarks. Furthermore, AutoL2S achieves nearly the same average accuracy as the oracle SFT R1-distilled models (i.e., Bespoke-Stratos-3B/7B), while significantly reducing reasoning path length. This presents the same observation showcased in Section 4.

Considering standard deviation, AutoL2S continues to outperform both the oracle SFT R1-distilled models and other baselines, offering better accuracy and lower average token usage. For example, with AutoL2S based on Qwen2.5-7B-Instruct, the performance remains the best among all methods, while also achieving the shortest reasoning lengths. These results demonstrate that AutoL2S has both competitive and robust performance in efficient reasoning tasks.

Table 5: Evaluation results of AutoL2S based on Qwen2.5-3B-Instruct.(mean ± std)

| | Average | | MATH500 | | GPQA | | GSM8K | | Olympiad | |
|---|---|---|---|---|---|---|---|---|---|---|
| | Acc | Len | Acc | Len | Acc | Len | Acc | Len | Acc | Len |
| Qwen2.5-3B-Instruct | 0.479 | 777 | 0.622 | 806 | 0.349 | 770 | 0.679 | 376 | 0.266 | 1158 |
| Bespoke-Stratos | 0.516 | 8931 | 0.636 | 9246 | 0.308 | 10129 | 0.848 | 1624 | 0.272 | 14724 |
| CoT-Valve | 0.484 | 5889 | 0.602 | 4980 | 0.258 | 6898 | 0.805 | 1660 | 0.270 | 10017 |
| O1-pruner | **0.535** | 6686 | **0.704** | 6769 | 0.283 | 7348 | **0.859** | 1210 | **0.295** | 11416 |
| AutoL2S (rj = 0) | 0.523 ±0.006 | 5083 ±737 | 0.656 ±0.015 | 4287 ±605 | 0.322 ±0.003 | **4018** ±941 | 0.830 ±0.026 | 1109 ±224 | 0.284 ±0.023 | 10919 ±1293 |
| AutoL2S (rj = 4) | 0.524 ±0.009 | 3569 ±506 | 0.646 ±0.016 | 2713 ±135 | **0.347** ±0.015 | 4118 ±514 | 0.826 ±0.003 | 503 ±4 | 0.278 ±0.007 | 6942 ±1915 |
| AutoL2S (rj = 8) | 0.523 ±0.007 | **3255** ±548 | 0.671 ±0.021 | 2523 ±200 | 0.317 ±0.008 | 4135 ±598 | 0.825 ±0.004 | **417** ±41 | 0.280 ±0.005 | **5947** ±1796 |

Table 6: Evaluation results of AutoL2S based on Qwen2.5-7B-Instruct.(mean ± std)

| | Average | | MATH500 | | GPQA | | GSM8K | | Olympiad | |
|---|---|---|---|---|---|---|---|---|---|---|
| | Acc | Len | Acc | Len | Acc | Len | Acc | Len | Acc | Len |
| Qwen2.5-7B-Instruct | 0.586 | 435 | 0.748 | 556 | 0.308 | 27 | 0.902 | 260 | 0.384 | 896 |
| Bespoke-Stratos | 0.638 | 6019 | 0.824 | 5383 | 0.359 | 6049 | 0.926 | 1321 | 0.444 | 11322 |
| CoT-Valve | 0.594 | 4747 | 0.730 | 4483 | 0.369 | 4930 | 0.898 | 928 | 0.378 | 8647 |
| O1-pruner | 0.650 | 5267 | **0.832** | 5104 | 0.399 | 5312 | **0.936** | 1065 | 0.433 | 9586 |
| AutoL2S (rj = 0) | **0.652** ±**0.007** | 4348 ±306 | 0.795 ±0.005 | 3278 ±240 | **0.431** ±**0.006** | 4590 ±532 | 0.923 ±0.011 | 595 ±150 | **0.460** ±**0.010** | 8932 ±335 |
| AutoL2S (rj = 4) | 0.630 ±0.011 | 3233 ±474 | 0.788 ±0.017 | 2200 ±354 | 0.375 ±0.033 | 3103 ±494 | 0.915 ±0.003 | 439 ±68 | 0.442 ±0.009 | 7190 ±994 |
| AutoL2S (rj = 8) | 0.626 ±0.013 | **2746** ±**496** | 0.785 ±0.012 | **2019** ±**368** | 0.380 ±0.019 | **2587** ±**799** | 0.915 ±0.015 | **415** ±**75** | 0.422 ±0.016 | **5964** ±**921** |

## K  CASE STUDIES OF AUTOL2S

In this section, we present additional case studies to demonstrate the efficiency of AutoL2S. The AutoL2S responses shown in this section are generated by a model trained on Qwen2.5-7B-Instruct with $rj = 8$. The baseline responses in the first two examples are produced by Bespoke-Stratos-7B (Labs, 2025). We include three types of inference modes: Auto Long Reasoning, Auto Short Reasoning, and AutoL2S without <EASY>.

### K.1  QUALITATIVE ANALYSIS OF AUTOL2S

The AutoL2S response shown here is generated by a model trained in Qwen2.5-7B-Instruct with $rj = 8$, and the baseline response is produced by Bespoke-Stratos-7B. The case studies demonstrate clear improvements in AutoL2S output after training. Before training, responses were often verbose, meandering, and filled with self-doubt, with frequent backtracking and confusion in applying domain knowledge. This will lead to several rounds of redundant thinking and verification steps. We observe that in some simple questions, complex verification steps can be reduced or even omitted, where LLMs are strong enough to precede the answers. After training with AutoL2S, the outputs became more structured and coherent, with reasoning expressed in a logical step-by-step manner and conclusions stated concisely and decisively. The use of special triggers aligned better with the intended long–short reasoning split, domain knowledge was applied more accurately, and verbose exploration was replaced by tighter yet sufficiently detailed explanations. Overall, training enhanced fluency, confidence, and correctness while reducing redundancy, showing a clear shift from rambling exploration to disciplined, solution-oriented reasoning.

### K.2  AUTO LONG REASONING FOR REGULAR QUESTIONS

We showcase an example when encountering a regular question. AutoL2S automatically chooses to reason longer but eliminates the token usage compared to baseline responses.

---

**Case Study: The Lollipop Packing Problem**

**User Prompt:** Jean has 30 lollipops. Jean eats 2 of the lollipops. With the remaining lollipops, Jean wants to package 2 lollipops in one bag. How many bags can Jean fill?

**Baseline Response (445 Tokens):**
<|begin_of_thought|> Okay, let's see. Jean has 30 lollipops. She eats 2 of them. Then she wants to put the remaining ones into bags, with each bag holding 2 lollipops. The question is, how many bags can she fill? Hmm, first step, I need to figure out how many lollipops Jean has left after she eats 2. So originally, she has 30. If she eats 2, that means she subtracts 2 from 30. Let me write that down: 30 minus 2 equals 28. So she has 28 lollipops left. Now, she wants to put 2 lollipops in each bag. So, to find out how many bags she can fill, I need to divide the number of remaining lollipops by 2. That would be ...[251 tokens omitted]

**AutoL2S Response (136 Tokens):**
<|begin_of_solution|> **<EASY><Short Trigger>** This is a trigger to ensure the model's upcoming output <short>. Let's approach this step by step. Jean starts with 30 lollipops. She eats 2, so we need to subtract those from the total. 30 minus 2 equals 28. So, Jean has 28 lollipops left. Now, Jean wants to put 2 lollipops in each bag. To find out how many bags she can fill, we divide the number of remaining lollipops by the number of lollipops per bag. 28 divided by 2 equals 14. Therefore, Jean can fill 14 bags. <|end_of_solution|>

---

---

**Case Study: Auto Long Reasoning for Regular Questions**

**User Prompt:** Melissa works as a pet groomer. This week, she has 8 dogs that need to be bathed, 5 cats that need their nails clipped, 3 birds that need their wings trimmed, and 12 horses that need to be brushed. If she splits the grooming jobs evenly over the days, how many animals will she groom each day of the week?

**Baseline Response (2207 Tokens):**

```
<|begin_of_thought|>
```
Okay, let's see. Melissa is a pet groomer, and she has different animals to groom this week. The problem is asking how many animals she'll groom each day if she splits the jobs evenly over the days. Alright, so first, I need to figure out the total number of animals she has to groom. Let me list them out:
- Dogs: 8 that need bathing.
- Cats: 5 that need their nails clipped.
- Birds: 3 that need their wings trimmed.
- Horses: 12 that need brushing.
...[1889 tokens omitted]...
```
<|end_of_thought|>
<|begin_of_solution|>
```
Melissa has a total of 8 dogs, 5 cats, 3 birds, and 12 horses to groom. To find the total number of animals:

$$8\,(\text{dogs}) + 5\,(\text{cats}) + 3\,(\text{birds}) + 12\,(\text{horses}) = 28\,(\text{total animals})$$

The problem asks how many animals she will groom each day if the jobs are split evenly over the days. Assuming she works 7 days a week, the calculation is:

$$\frac{28\,(\text{total animals})}{7\,(\text{days})} = 4\,(\text{animals per day})$$

Thus, the final answer is $\boxed{4}$.

```
<|end_of_solution|>
```

**AutoL2S Response (1351 Tokens):**

```
<|begin_of_thought|>
```
Let's consider this problem in a `<pureLong>` way. Okay, let's see. So the problem is about Melissa, who's a pet groomer. She has different animals to groom: 8 dogs, 5 cats, 3 birds, and 12 horses. The question is asking how many animals she will groom each day if she splits the jobs evenly over the days of the week.
First, I need to figure out the total number of grooming jobs she has.
...[1112 tokens omitted]...
```
<|end_of_thought|>
<|begin_of_solution|>
```
Melissa has a total of 8 dogs, 5 cats, 3 birds, and 12 horses to groom. Adding these together gives:

$$8 + 5 + 3 + 12 = 28$$

She needs to split these 28 grooming jobs evenly over 7 days of the week. Dividing the total number of jobs by the number of days gives:

$$28 \div 7 = 4$$

Thus, the number of animals she will groom each day is:

$$\boxed{4}$$

```
<|end_of_solution|>
```

### K.3  AUTO SHORT REASONING FOR EASY QUESTIONS

We showcase the example when AutoL2S encounter a regular question. The results show AutoL2S can automatically switch to from long reasoning path to short reasoning path without sacrificing the reasoning performance.

---

**Case Study: Auto Short Reasoning for EASY Questions**

**User Prompt:** Andrew plans a road trip from New Jersey to Rochester. It takes 6 days to travel by bus and half as many days to travel by car from New Jersey to Rochester. How many days will he travel if he plans to go to Rochester by bus and return from Rochester by car?

**Baseline Answer (573 Tokens):**

`<|begin_of_thought|>`

Okay, so Andrew is planning a road trip from New Jersey to Rochester. The problem says it takes 6 days to travel by bus, and half as many days by car. Then he wants to go by bus and return by car. I need to figure out the total number of days he'll travel.

First, let me parse the information. The trip from New Jersey to Rochester by bus takes 6 days. Then, the return trip by car would take half as many days as the bus trip. So, half of 6 days is 3 days. That makes sense because the problem states "half as many days to travel by car," so it's half the bus time.

So, going by bus: 6 days. Coming back by car: 3 days. To find the total travel time, I need to add both legs of the journey. That would be 6 days + 3 days. Let me do the math: 6 + 3 equals 9. So, the total travel time is 9 days.

Wait a second, let me make sure I didn't miss anything. The problem says he plans to go to Rochester by bus and return from Rochester by car. So, it's a round trip. The key here is that the return trip is from Rochester back to where he started, which is New Jersey. So, the distances are the same, but the mode of transportation changes.

The bus trip to Rochester is 6 days, car trip back is half that, so 3 days. Adding them together gives 9 days total travel time. That seems straightforward. Is there any part of this that could be interpreted differently?

For example, maybe the problem is referring to something else, like including time in New Jersey or Rochester? But the problem specifically mentions the travel time by bus and car for the same distance. So, it's just the round trip, with each leg's time as given.

Another check: if the bus takes 6 days, then half of that is indeed 3 days. So, 6 + 3 is 9. Yeah, that seems right. I don't see any other factors here. The answer should be 9 days.

`<|end_of_thought|>`

`<|begin_of_solution|>`

Andrew plans to travel from New Jersey to Rochester by bus, which takes 6 days. The return trip by car takes half as many days as the bus trip, which is $6 \div 2 = 3$ days. The total travel time is the sum of both trips:

$$6 \text{ days (bus)} + 3 \text{ days (car)} = 9 \text{ days}$$

The final answer is $\boxed{9}$.

`<|end_of_solution|>`

**AutoL2S Answer (200 Tokens):** `<|begin_of_solution|>`

This is a trigger to ensure the model's upcoming output `<short>`. To determine the total number of days Andrew will travel, we need to consider both the trip to Rochester and the return trip from Rochester. Let's break it down step by step.

1. **Trip to Rochester by bus:** - It takes 6 days to travel by bus from New Jersey to Rochester.
2. **Return trip from Rochester by car:** - The problem states that it takes half as many days to travel by car as it does by bus. - Since the bus trip takes 6 days, the car trip will take $\frac{6}{2} = 3$ days.
3. **Total travel time:** - Add the time taken for both trips: 6 days (bus)$+3$ days (car) $= 9$ days.

Therefore, the total number of days Andrew will travel is $\boxed{9}$.

`<|end_of_solution|>`

---

## K.4 AUTOL2S W/O <EASY>

We showcase an example of not using <EASY> in AutoL2S.

---

**Case Study: AutoL2S w/o <EASY>**

**User Prompt:** Janet's ducks lay 16 eggs per day. She eats three for breakfast every morning and bakes muffins for her friends every day with four. She sells the remainder at the farmers' market daily for $2 per fresh duck egg. How much in dollars does she make every day at the farmers' market?

**AutoL2S w/o <EASY> (986 Tokens):**

```
<|begin_of_thought|>
```
<specialLong> We will provide a detailed explanation and solution. Okay, let's see. So Janet's ducks lay 16 eggs every day. That's the starting point. Then she does a couple of things with those eggs. First, she eats three for breakfast every morning, and then she uses four to bake muffins for her friends each day. The rest of the eggs, which are left after she uses them for her own consumption and baking, she sells at the farmers' market for $2 each. The question is asking how much money she makes from selling those eggs each day.

...[504 tokens omitted]...

```
<|end_of_thought|>
<|begin_of_solution|>
```
Janet starts with 16 eggs each day. She eats 3 eggs for breakfast and uses 4 eggs to bake muffins, totaling $3 + 4 = 7$ eggs used per day. Subtracting this from the daily gives $16 - 7 = 9$ eggs left to sell. Each egg is sold for $2, so the total money made from selling the eggs is $9 \times 2 = 18$ dollars. The final answer is $\boxed{18}$.`<|end_of_solution|><seperate><|begin_of_solution|>`
This is a trigger to ensure the model's upcoming output <short>. To determine how much Janet makes every day at the farmers' market, we need to follow these steps:

1. **Calculate the total number of eggs laid by the ducks per day:** Janet's ducks lay 16 eggs per day.

2. **Determine the number of eggs Janet eats for breakfast each day:** Janet eats 3 eggs for breakfast every morning.

3. **Determine the number of eggs Janet uses to bake muffins each day:** Janet bakes muffins for her friends every day with 4 eggs.

4. **Calculate the total number of eggs Janet uses each day:**

$$3 \text{ (eggs for breakfast)} + 4 \text{ (eggs for muffins)} = 7 \text{ eggs}$$

5. **Calculate the number of eggs remaining after Janet uses some for breakfast and baking:**

$$16 \text{ (total eggs)} - 7 \text{ (eggs used)} = 9 \text{ eggs}$$

6. **Determine the amount of money Janet makes from selling the remaining eggs:** Janet sells the remaining 9 eggs at the farmers' market for $2 per egg.

$$9 \text{ eggs} \times 2 \text{ dollars per egg} = 18 \text{ dollars}$$

Therefore, the amount of money Janet makes every day at the farmers' market is $\boxed{18}$.

```
<|end_of_solution|>
```
**AutoL2S w <EASY> (314 Tokens):**

```
<|begin_of_solution|>
```
This is a trigger to ensure the model's upcoming output <short>. To determine how much Janet makes every day at the farmers' market, we need to follow these steps:

1. **Calculate the total number of eggs laid by the ducks per day:** Janet's ducks lay 16 eggs per day. ...[158 tokens omitted]...

4. **Determine how much Janet makes from selling the remaining eggs:** Janet sells the remaining 9 eggs at the farmers' market for $2 per egg. Therefore, the total amount she makes is:

$$9 \text{ eggs} \times 2 \text{ dollars per egg} = 18 \text{ dollars}$$

Thus, the amount Janet makes every day at the farmers' market is $\boxed{18}$ dollars.

```
<|end_of_solution|>
```

---

## L   VISUALIZATIONS FOR AUTOL2S' OPTIMIZATION TRACE

Figure 3 visualizes the optimization trajectory of AutoL2S and AutoL2S-Plus on Qwen2.5–7B–Instruct model, illustrating how they achieve the trade-off between reasoning length (Avg. # Tokens) and accuracy (Avg. Accuracy). Specifically, AutoL2S employs long-short distillation to move beyond the instruct model toward higher accuracy, achieving better performance and shorter reasoning paths than the long-only distilled model. Building upon this, AutoL2S-Plus further compresses the reasoning length without compromising accuracy by optimizing under the length-aware reinforcement learning objective. These optimization traces demonstrate that, by dynamically balancing long and short reasoning paths, AutoL2S and AutoL2S-Plus achieves efficient reasoning—substantially reducing reasoning length while preserving task accuracy.

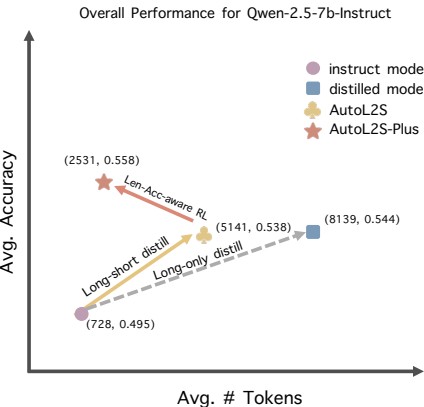

Figure 3: Visualizations for AutoL2S' optimization trace.

## M   PARETO FRONT OF REASONING ACCURACY AND EFFICIENCY

In this section, we showcase the Pareto Front of different methods in terms of their wall-clock time and KV cache peak memory. The results are shown in Figure 4. We observe that AutoL2S obtains the best trade-off between accuracy and reasoning efficiency.

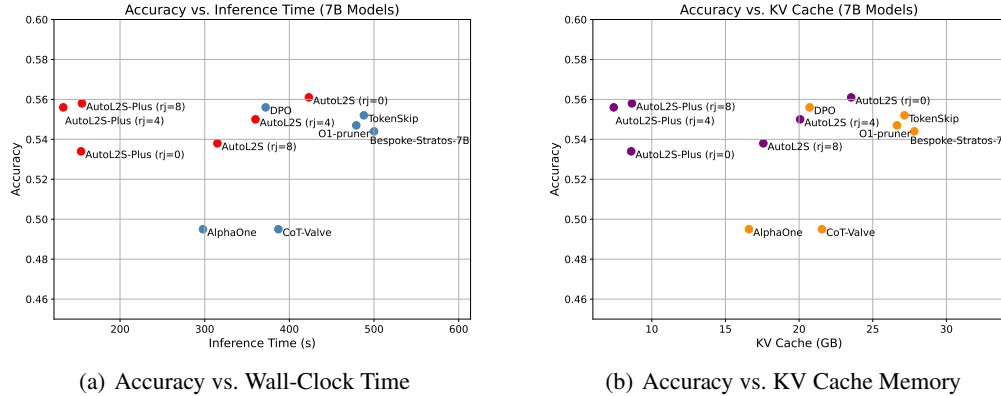

(a) Accuracy vs. Wall-Clock Time        (b) Accuracy vs. KV Cache Memory

Figure 4: Efficiency–accuracy trade-off of 7 B models. Left: wall-clock inference time; Right: KV-cache memory footprint. AutoL2S variants consistently improve accuracy under substantially lower inference cost.

