# OpenReview forum: "AutoL2S: Auto Long-Short Reasoning for Efficient Large Language Models"
_ICLR.cc/2026/Conference — ICLR 2026 Conference Withdrawn Submission_

### Official Review · Reviewer_m2zH · 2025-10-30

**Soundness:** 2
**Presentation:** 2
**Contribution:** 1
**Rating:** 2
**Confidence:** 4

**Summary:**

This paper targets the "overthinking" problem in LLMs, wherein reasoning-capable models generate excessively long CoT paths even for simple queries. The authors propose Auto Long-Short Reasoning (AutoL2S), described as a dynamic and model-agnostic framework. The approach involves fine-tuning a non-reasoning LLM on a specially prepared dataset that combines both long, full-length CoT paths and corresponding short, concise CoT paths. The core mechanism is the introduction of a special \<EASY\> token. During training, this token is associated with questions deemed solvable via a short path. During inference, the model is expected to autonomously generate this token upon identifying a simple question, which then signals the model to produce a concise answer and "skip generating lengthy CoT reasoning".

**Strengths:**

1. The paper addresses a problem of high practical importance. The "overthinking" phenomenon is a well-recognized bottleneck for the deployment of CoT-enabled LLMs, making research into reasoning efficiency both timely and valuable.
2. The central mechanism—using a learned, special token (\<EASY\>) as an explicit gate for dynamically controlling reasoning length—is intuitive and interpretable. It presents a potentially simpler alternative to complex reinforcement learning reward-shaping or training-free, inference-time monitoring systems.
3. The ablation studies on data annotation strategy (Table 3)  and the impact of the \<EASY\> token (Table 4)  are useful. The comparison between the "w/ Force-Long" setting and the full AutoL2S model provides evidence supporting the efficacy of the dynamic switching mechanism itself.

**Weaknesses:**

1. The paper's primary weakness is its failure to properly situate its contribution within the existing literature. The central problem of "overthinking"  and the proposed solution—a SFT framework to enable self-regulating reasoning length—are not novel. The paper omits citation of its most direct competitors, like Self-Braking Tuning (SBT), making it impossible to assess its incremental contribution. Both AutoL2S and SBT are SFT-based frameworks. Where AutoL2S uses an \<EASY\> token , SBT employs "natural language braking signals". This appears to be a minor implementation difference rather than a fundamental conceptual one.

2. The paper's two theorems, intended to provide formal backing, are weak and add little scientific substance.
- Theorem 1 is a restatement of an information-theoretic truism. The proof 1 rests entirely on the fact that "conditioning reduces entropy," i.e., $H(S_{t} | X, L, S_{<t}) \le H(S_{t} | X, S_{<t})$. It is obvious that the long path $L$ provides additional mutual information about the short path $S$, as they are both solutions to the same problem $X$. This theorem does not prove that a neural network trained on this concatenated sequence learns more effectively, generalizes better, or converges faster than a model trained on $S$ alone or on $L$ and $S$ as separate data points.
- Theorem 2 serves as a post-hoc justification that is disconnected from the actual training mechanism. The paper defines a risk function (Equations 8 and 9) and then claims—without proof—that the standard SFT cross-entropy objective (Equation 1) "implicitly minimizes the per-instance risk". There is no demonstrated link between the two. The terms in the risk function (e.g., statistical divergence $D(\cdot||\cdot)$, per-token cost $\lambda$, overhead $c_{\pi}$) 1 have no corresponding terms in the SFT loss function. The model learns to predict \<EASY\> simply because it is performing pattern matching on the training data, which explicitly includes this token for "easy" questions. It is not "approximating an optimal adaptation policy."

3. The experimental setup contains a major confounding variable that makes attribution of the paper's positive results difficult. The "short CoT reasoning paths" are not generated by the same teacher model (DeepSeek-R1) or by compressing its outputs. Instead, they are generated by a completely different, math-specialized model: Qwen2.5-Math-7B-Instruct. This design choice introduces a massive confound. The paper claims to be testing the AutoL2S framework (i.e., joint training with an \<EASY\> token). However, the experiment also implicitly tests the effect of distilling knowledge from Qwen2.5-Math into the base models. The Qwen2.5-Math model is noted as having "inherent shorter reasoning path[s]". Therefore, the observed efficiency gains (shorter tokens) may not be due to the AutoL2S mechanism at all, but rather due to the student model simply learning the more concise reasoning style of the Qwen2.5-Math teacher.

**Questions:**

See weaknesses

---

> ### Author Response · Authors · 2025-11-19
> **Part 1/3**
>
> #### **[Q-1]** Both AutoL2S and SBT are SFT-based frameworks. Where AutoL2S uses an \<EASY\> token , SBT employs "natural language braking signals". This appears to be a minor implementation difference rather than a fundamental conceptual one.
>
> **[A-1]**
> We respectfully disagree that the difference between AutoL2S and SBT is only an implementation detail. The two frameworks differ in both training paradigm and overthinking identification.
>
> **(1) Training Paradigm: SFT vs. SFT → Off-Policy RL**
> SBT is a pure SFT-based method to prevent overthinking. In contrast, AutoL2S is a **two-stage SFT-then-off-policy RL** framework:
>
> * **Stage 1: Long–short concatenated distillation (SFT).** We obtain a long–short aligned model capable of automatically switching CoT modes.
> * **Stage 2: Off-policy RL with a length-aware objective.** We further push the accuracy–length Pareto frontier by rewarding generations that shorten reasoning while preserving correctness (Sec. 3.4). Since the length objective is non-differentiable, we optimize it with PPO-style clipped loss and compute per-sample advantage by leveraging the long- and short- form outputs of the sft-based AutoL2S model which is used as reference policy.
>
> Anchoring RL on a long–short SFT model as reference and base policy is new relative to SBT and empirically critical for further stable length control and accuracy preservation. As shown in Table1 and Fig. 3 in Appendix L, RL training (avg. acc.0.558, avg. len. 2531) achieves a **68.9% reduction in reasoning length** while improving accuracy by **+0.014**, further improving the accuracy–length trade-off beyond the SFT-based AutoL2S(avg acc 0.538;avg len 5141). Compared to RL-only training(avg. acc. 0.547, avg. len. 7797), i.e. O1-pruner, AutoL2S-Plus both **shortens the reasoning path significantly and maintains higher accuracy**, demonstrating the benefit of leveraging **a stronger reference model (AutoL2S)** rather than optimizing directly from the original distilled policy (i.e., Bespoke-Stratos-7B).
>
> **(2) Overthinking Identification: stepwise pruning vs. one-shot selection**
> SBT dynamically compresses intermediate thoughts and requires **heuristic scoring functions** to prune reasoning tokens **at each generation step**. AutoL2S follows a different principle: it does **not compress or edit intermediate thoughts**. Instead, it makes **a single decision** based the correctness and reasoning length. This avoids complex heuristic ovethinking socre design and is substantially more **efficient** than stepwise inspection.
>
> **(3) Long–short concatenated distillation VS Masked redundant thinking finetuning**
> AutoL2S introduces a supervision signal where short CoT is trained conditioned on the full long CoT. We use a long-short concatenation format with an explicit switch token, where the long trace serves as an informative prefix rather than material to be compressed. This conditional long-short concatenation for supervision signal is different from SBT's masked redundant thinking finetuning, which masks redundent steps within a single long CoT.
>
> In summary, AutoL2S and SBT share a high-level goal of mitigating overthinking, but they differ fundamentally in (i) training paradigm—SFT-only vs. SFT → Off-Policy RL with a length-aware objective, and (ii) overthinking detection—stepwise heuristic compression vs. one-shot CoT selection (iii) long–short concatenated distillation VS masked redundant thinking finetuning. AutoL2S therefore represents a distinct and more general approach rather than a minor implementation variant.
>
> [1] Zhao, Haoran, et al. "Let LLMs Break Free from Overthinking via Self-Braking Tuning." arXiv preprint arXiv:2505.14604 (2025).

---

> > ### Author Response · Authors · 2025-11-19
> > **Part 2/3**
> >
> > #### **[Q-2]** The paper's two theorems, intended to provide formal backing, are weak and add little scientific substance.
> >
> > **[A-2]**
> >
> > We thank the reviewer for the comment. We would like to clarify that we do **not** present it as a new theoretical result, instead we aim to provide the theoretical explanation to interpret the empirical findings.
> >
> > - Theorem 1 indeed relies on the classical information-theoretic fact that conditioning reduces entropy, and we do **not** present it as a new theoretical result. Its purpose is to support and interpret the empirical findings. Formally, Theorem 1 states that the long CoT path (L) provides additional mutual information to the short path (S):$I(S_t; L \mid X, S_{<t}) \ge 0$, which implies:$H(S_t \mid X, L, S_{<t}) \le H(S_t \mid X, S_{<t}).$ This analytical framing explains why the model benefits from the concatenated long–short training structure and aligns with the empirical behavior observed in Sec. 4.6 and Fig. 2, where attention becomes more sparse and focused. We will revise the paper to explicitly state that the theorem is an analytical explanation supporting the empirical results rather than a new theoretical contribution.
> > - We would like to clarify that the reviewer’s interpretation of Theorem 2 does not accurately reflect its intended role in the paper. While we agree on the technical point that Theorem 2 is **not a derivation of the SFT loss (Eq. 1)**, this is by design rather than by omission. The theorem is not meant to characterize the training objective; instead, it provides a **decision-theoretic explanation of the model’s empirical inference-time behavior**, which is a central component of AutoL2S. AutoL2S explicitly trains the model to output either `<EASY>` or `<Long Trigger>` depending on input difficulty. After training, the model consistently exhibits an adaptive selection pattern: The model tends to choose the short reasoning path when the predictive distributions are close: $p_S(\cdot \mid X) \approx p_L(\cdot \mid X)$, and defaults to long reasoning otherwise. This behavior is consistent with the optimal rule: $\text{choose short if } J_S(X) < J_L(X).$ Thus, Theorem 2 is not “disconnected”; it is purposefully aligned with the empirical phenomenon that motivates AutoL2S. It serves as an analytical framework that explains why the model behaves as it does, rather than as a derivation of the SFT loss. We will revise the text to make this interpretive role explicit and remove any phrasing that might be misread as implying objective-level equivalence.

---

> > > ### Author Response · Authors · 2025-11-19
> > > **Part 3/3**
> > >
> > > #### **[Q-3]** The experimental setup contains a major confounding variable that makes attribution of the papers positive results difficult. The short CoT reasoning paths are not generated by the same teacher model (DeepSeek-R1) or by compressing its outputs. Instead, they are generated by a completely different, math-specialized model: Qwen2.5-Math-7B-Instruct. This design choice introduces a massive confound. The paper claims to be testing the AutoL2S framework (i.e., joint training with an EASY token). However, the experiment also implicitly tests the effect of distilling knowledge from Qwen2.5-Math into the base models. The Qwen2.5-Math model is noted as having inherent shorter reasoning path[s]. Therefore, the observed efficiency gains (shorter tokens) may not be due to the AutoL2S mechanism at all, but rather due to the student model simply learning the more concise reasoning style of the Qwen2.5-Math teacher.
> > >
> > > **[A-3]**
> > > We appreciate the reviewer’s concern, but the efficiency gains of AutoL2S **cannot** be explained by Qwen2.5-Math distillation alone. Our design explicitly controls for this, and the AutoL2S mechanism, rather than teacher identity, is what drives reasoning efficiency.
> > >
> > > (1) The goal of AutoL2S is to distill both long and short reasoning so the model learns when to use each mode based on the EASY tokens. The short traces are not meant to impose the teacher’s writing style. What matters is having a paired long–short supervision signal, not which model produced the short trace. After the SFT training, we leverage off-policy RL to further push the Pareto front of accuracy and reasoning length. Both steps in AutoL2S contribute to efficiency enhancement.
> > >
> > > (2) Section 4.4 and Table 3 directly test the “different-teacher” hypothesis. When we train only on long CoT from DeepSeek-R1—no short traces, no Qwen2.5-Math, the model does not become shorter. If the improvements came simply from mixing in a math teacher, this control would show similar benefits. It doesn’t. This comparison makes it clear that using Qwen2.5-Math data is not the only driving factor. Furthermore, in Table 4, AutoL2S outperforms Qwen2.5-Math-7B itself in terms of getting a better trade-off between accuracy and reasoning length.
> > >
> > > (3) The shortening effect comes from the mechanisms we introduce: the \<EASY\> token and the RL objective given the SFT model as a reference model. The token gives the model a clear gating signal, and the RL loss encourages it to balance accuracy with conciseness. Using AutoL2S as the reference policy further ensures the optimization tracks the divergence between long and short traces, rather than any stylistic preferences of Qwen2.5-Math.
> > >
> > > Taken together, these elements form a coherent system:
> > > AutoL2S learns to shorten its reasoning only when short reasoning is sufficient, not because it is copying the short-trace teacher.

---

> > > > ### Comment · Reviewer_m2zH · 2025-11-27
> > > > **Thanks to authors' detailed responses**
> > > >
> > > > After reading the author's responses to me and the other reviewers, most of my technical concerns have been addressed. Therefore, I decided to raise my score by two points.
> > > >
> > > > However, as several other reviewers have pointed out, the current method seems to lack a key distinguishing feature compared to many previous methods (it might be due to the writing).

---

> ### Author Response · Authors · 2025-11-28
>
> We sincerely thank the reviewer for the follow-up and for recognizing that most technical concerns have been addressed. We believe that the concerns and misunderstandings raised can be cleared up and clarified by making simple expository improvements.
>
> To clarify the distinguishing features of AutoL2S, we provide a concise comparison below, highlighting how our framework differs from prior methods both conceptually and technically.
>
> **1.Long–short concatenated distillation**
> Prior methods (TokenSkip[1], CoT-Valve[2], DPO) mainly prune redundancy or rank preferences. AutoL2S instead performs conditional long→short distillation: short CoT is trained with the full long trace as an informative prefix. This supervision, teaching the model to synthesize concise reasoning from full traces, does not exist in prior work and goes beyond mere filtering or ranking.
>
> **2. Implicit  \<EASY\> gate from paired supervision**
> Where previous routing methods require explicit classifiers or handcrafted rules, AutoL2S learns routing directly from paired long/short data. The  \<EASY\> token naturally emerges as a data-driven threshold, enabling routing without extra modules or parameters.
>
> **3. SFT then off-policy RL with the AutoL2S SFT model as reference and base policy**
> Unlike length-aware RL baselines (e.g., O1-Pruner[3]) that apply RL on a fixed reasoning-capable teacher, AutoL2S first builds a long–short-aligned SFT model that already switches between long/short modes, and then uses this SFT model as both base and reference in off-policy RL. This design is new and empirically crucial for stable length control and accuracy preservation (Tables 1, 3, 4, Figure 4).
>
> |AutoL2S|Prior|Key Delta|
> |-|-|-|
> |Conditional long→short distillation|Token-level Pruning/Compression| Short CoT is trained *conditioned on full long CoT* (not truncation or ranking).|
> |Implicit \<EASY\> gate|Extra gating/controllers|Gate is *implicitly learned*, no classifier/router.|
> |SFT then off-policy RL with AutoL2S reference|Length-aware RL|AutoL2S uses a long–short SFT as base and reference policy.|
>
> We will make these distinctions clearer in the revised version to ensure the novelty is more evident in writing. We truly appreciate your constructive feedback and hope this clarification further strengthens your confidence in recommending acceptance.
>
>
> [1] Xia, Heming, et al. "Tokenskip: Controllable chain-of-thought compression in llms." arXiv preprint arXiv:2502.12067 (2025).
>
> [2] Luo, Haotian, et al. "O1-pruner: Length-harmonizing fine-tuning for o1-like reasoning pruning." arXiv preprint arXiv:2501.12570 (2025).
>
> [3] Ma, Xinyin, et al. "Cot-valve: Length-compressible chain-of-thought tuning." arXiv preprint arXiv:2502.09601 (2025).

---

### Official Review · Reviewer_ett5 · 2025-10-31

**Soundness:** 2
**Presentation:** 3
**Contribution:** 3
**Rating:** 4
**Confidence:** 4

**Summary:**

This paper presents AutoL2S to stop AI models from overthinking simple questions. The solution is to train the model on a new dataset that includes both long, detailed answers and correct, short answers. A special <EASY> token is added to the training data for simple questions, teaching the model how to recognize them. As a result, when the model is asked a new question, it first decides if it's easy or hard; if it's easy, it uses the <EASY> token to give a quick, short answer, but if it's hard, it uses the full, long reasoning path. This makes the model much more efficient, cutting its answer length by more than half without losing accuracy.

**Strengths:**

1. Instead of complex dataset rebuilding, this paper introduces a much easier way for constructing the datasets. The paper validate the method across multiple model families and diverse reasoning benchmarks, and provide extensive ablation studies and mechanistic analyses to prove why it works.

2. It achieves a over 50% reduction in reasoning length while maintaining or even improving accuracy provides a direct, high-impact solution for building faster, cheaper, and more scalable reasoning applications.

**Weaknesses:**

1. The paper's evaluation is narrowly focused on maths and physics benchmarks. It makes unclear if the AutoL2S framework's ability to distinguish "easy" from "hard" questions will successfully generalize to other complex reasoning domains, such as commonsense, or coding. **OOD experiments are required to show the performance really does not "drop".**

2. There is no guidance on when this method works well. Later, if readers have their own datasets and domains, under what conditions will this method still perform effectively? For example, in Table 1, after training, the performance on GPQA increases while others drop. Why does it increase?

3. All the accuracy except GPQA drops, so in order to save tokens, will methods which dynamically compress intermediate thoughts will work better than this method? The ultimate goal is the same.

**Questions:**

1. compared to papers like "LightThinker", what is the advantage? (https://arxiv.org/pdf/2502.15589)

2. The Table 1 is so hard to read. Bold or colorbox might help.

3. Can you run experiments on OOD datasets, such as coding or BBH?

---

> ### Author Response · Authors · 2025-11-19
> **Part 1/2**
>
> #### **[Q-1]** The paper's evaluation is narrowly focused on maths and physics benchmarks. It makes unclear if the AutoL2S framework's ability to distinguish "easy" from "hard" questions will successfully generalize to other complex reasoning domains, such as commonsense, or coding. OOD experiments are required to show the performance really does not "drop". Can you run experiments on OOD datasets, such as coding or BBH?
>
> **[A-1]**
> We would like to emphasize that all experimental settings follow prior work [1, 2, 3], and the math dataset is widely adopted as a standard benchmark [4] for evaluating LLM reasoning performance. To further address the reviewer’s concerns, we also conduct experiments on the LiveCodeBench v2 dataset[5], a widely used coding reasoning benchmark. As shown in the table, AutoL2S consistently outperforms Bespoke-Stratos-7B in both accuracy and output length.
>
> |Model|Acc|Len|
> |-|-|-|
> |Bespoke-Stratos-7B|0.308|19200|
> |AutoL2S(rj=0)|0.327|16907|
> |ΔvsBespoke|**+6.2%**|**–11.9%**|
> |AutoL2S(rj=4)|0.3654|15681|
> |ΔvsBespoke|**+18.6%**|**–18.3%**|
> |AutoL2S(rj=8)|0.346|15295|
> |ΔvsBespoke|**+12.3%**|**–20.3%**|
> |AutoL2S-Plus|0.327|15295|
> |ΔvsBespoke|**+6.17%**|**–42.9%**|
>
> [1] Wang, et al. "Thinking preference optimization." arXiv preprint arXiv:2502.13173 (2025).
>
> [2] Luo, et al. "O1-pruner: Length-harmonizing fine-tuning for o1-like reasoning pruning." arXiv preprint arXiv:2501.12570 (2025).
>
> [3] Ma, et al. "Cot-valve: Length-compressible chain-of-thought tuning." arXiv preprint arXiv:2502.09601 (2025).
>
> [4] Sui, et al. "Stop overthinking: A survey on efficient reasoning for large language models." arXiv preprint arXiv:2503.16419 (2025).
>
> [5] Jain, Naman, et al. "Livecodebench: Holistic and contamination free evaluation of large language models for code." arXiv preprint arXiv:2403.07974 (2024).
>
> #### **[Q-2]** There is no guidance on when this method works well. Later, if readers have their own datasets and domains, under what conditions will this method still perform effectively? For example, in Table 1, after training, the performance on GPQA increases while others drop. Why does it increase?
>
> **[A-2]**
> We would like to clarify that AutoL2S is a domain-agnostic and dataset-agnostic framework. Its dynamic switching between short and long reasoning paths is applicable to any task where reasoning redundancy exists—i.e., a subset of queries can be solved reliably with short CoT, while other queries truly require long-form reasoning. In such settings, AutoL2S aims to improve the accuracy–length Pareto frontier by reducing unnecessary reasoning while maintaining accuracy.
>
> In this paper, our experiments primarily focus on math and physics reasoning, and we additionally include a coding reasoning benchmark to demonstrate generality as shown in table1 under Q1. For new or domain-specific datasets, users may further improve performance by fine-tuning the model on an in-domain dataset to better adapt the framework.
>
> Regarding the performance drop, We would like to clarify that our goal is to **preserve the accuracy of the R1-distilled baseline while shortening the reasoning path**, rather than aiming to further improve accuracy. While it is always possible to achieve higher accuracy using longer CoT, or to shorten CoT length at the expense of bad accuracy, our focus is on optimizing the **trade-off between reasoning efficiency and accuracy preservation**, not on improving accuracy by sacrificing reasoning efficiency through longer CoTs.
>
> Experimental results support this objective. As shown in Table 1, under the 7B model setting, AutoL2S(rj=8) achieves a 36.8% reduction in token length while maintaining accuracy comparable to the oracle model Bespoke-Stratos-7B. In addition, compared with other baselines, AutoL2S achieves at least an 11.3\%–30\% reduction in token length with comparable performance.

---

> > ### Author Response · Authors · 2025-11-19
> > **Part 2/2**
> >
> > #### **[Q-3]** Will methods which dynamically compress intermediate thoughts will work better than this method? The ultimate goal is the same.
> >
> > **[A-3]**
> > Dynamic compression of intermediate thoughts is indeed another line of research for improving reasoning efficiency. Prior work [1, 2] typically performs **token-level or thinking-step-level compression** to shorten CoT reasoning paths. However, these approaches often require predefined scoring functions or manually designed hyperparameters to decide which parts of the reasoning are redundant. In contrast, AutoL2S does not modify or compress any intermediate reasoning tokens. Instead, it selects the shortest correct reasoning path from complete candidate CoT outputs, which **avoids potential semantic loss in token-level compression**. Moreover, compared to thinking-step-level approaches that must inspect and prune reasoning at every generation step, AutoL2S **makes a single early decision** based on the \<EASY\> trigger and then executes either a short or long CoT mode, making it **more efficient**.
> >
> > **Comparison to Dynamic Token-Level Compression**: One of our baseline methods tokenskip[1] compresses the intermediate CoT responses into shorter ones based on semantic scores, which is in the kind of Dynamically compressing the intermediate thought. As shown in Table1, under the Qwen2.5-7B-Instruct setting, AutoL2S (rj=4) achieves a 24% shorter response than TokenSkip (–27.8% vs –2.4%) while maintaining comparable accuracy (0.550 vs 0.552).
> >
> > **Comparison to Dynamic Thinking-Step-Level Compression**: We also compare AutoL2S with AlphaOne [2], which dynamically schedules slow–fast reasoning transitions via a Bernoulli process. As shown in the table below, AutoL2S achieves a 36.8% reduction in length with accuracy comparable to Bespoke-Stratos-7B, and AutoL2S-Plus achieves a 68.9% reduction in length while improving accuracy by +0.014. In contrast, AlphaOne reduces length by 40.3% but suffers a significant accuracy drop of -0.049.
> >
> > These experiment results demonstrate that AutoL2S achieves a superior accuracy–efficiency trade-off compared to dynamic intermediate-thought compression methods.
> >
> > |Model|acc|#tokens|MATH500|MATH500_tok|GPQA|GPQA_tok|GSM8K|GSM8K_tok|Olympiad|Olympiad_tok|AIME25|AIME25_tok|
> > |-|-|-:|-:|-:|-:|-:|-:|-:|-:|-:|-:|-:|
> > |Bespoke-Stratos-7B|0.544|8139|0.824|5383|0.359|6049|0.926|1321|0.444|11322|0.167|16619|
> > |AlphaOne|0.495|4856|0.732|3867|0.313|6278|0.907|1943|0.356|5252|0.167|6940|
> > |ΔvsBespoke|-0.049|-40.3%|-16.0%|-28.2%|-16.3%|+3.8%|-2.0%|+47.1%|-19.8%|-53.6%|+0.2%|-58.2%|
> > |AutoL2S(rj=8)|0.538|5141|0.798|2416|0.394|3492|0.929|488|0.436|6459|0.133|12852|
> > |ΔvsBespoke|-0.006|-36.8%|-3.2%|-55.1%|+9.7%|-42.3%|+0.3%|-63.1%|-1.8%|-43.0%|-20.0%|-22.7%|
> > |AutoL2S-Plus|0.558|2531|0.820|1719|0.424|3485|0.920|880|0.424|3041|0.200|3528|
> > |ΔvsBespoke|+0.014|-68.9%|-0.5%|-68.1%|+18.2%|-42.4%|-0.7%|-33.4%|-4.4%|-73.1%|+20.0%|-78.8%|
> >
> >
> >
> > #### **[Q-4]** Compared to papers like "LightThinker", what is the advantage?
> >
> > **[A-4]**
> > Thank you for the suggestion. LightThinker was preceding (September 23, 2025) after ICLR submission, so it could not be included in the original comparison based on the regulations of ICLR. However, we respect the comments from the reviewer. For completeness and to facilitate direct comparison to LightThinker, we incorporate quantitative results against LightThinker on GSM8K under Qwen-Distilled-R1-7B in the following table (i.e., we include the best results from LightThinker). The results demonstrate that AutoL2S outperforms LightThinker with the same accuracy but 50% shorter reasoning length. This means that AutoL2S can better push the trade-off between accuracy and reasoning length.
> >
> > ||Acc|Length|
> > |-|-|-|
> > |Distilled-R1|0.818|844|
> > |LightThinker|**0.871**|676|
> > |AutoL2S|**0.871**|**478**|
> >
> > [1] Zhang, Jintian, et al. "Lightthinker: Thinking step-by-step compression." arXiv preprint arXiv:2502.15589 (2025).
> >
> > #### **[Q-5]** The Table 1 is so hard to read. Bold or colorbox might help.
> >
> > **[A-5]** Thank you for the suggestions. We have updated Table 1 in the uploaded revision.

---

### Official Review · Reviewer_AVow · 2025-11-01

**Soundness:** 2
**Presentation:** 3
**Contribution:** 2
**Rating:** 4
**Confidence:** 3

**Summary:**

This paper proposes AutoL2S, a framework that enables LLMs to adjust reasoning effort according to question complexity. By jointly training on both long and short cot data and introducing an <EASY> token to skip unnecessary reasoning, AutoL2S improves efficiency with a small performance drop. The extended AutoL2S-Plus further calibrates reasoning length through length-aware fine-tuning. Experiments demonstrate up to 57% shorter reasoning with little accuracy loss.

**Strengths:**

1. AutoL2S can automatically decide between short and long reasoning based on the difficulty of the problem.
2. AutoL2S effectively reduces the reasoning length while maintaining only a small loss in accuracy.
3. The experimental results of AutoL2S appear to be effective and convincing.

**Weaknesses:**

1. According to Equation (4), for <EASY> questions, both long and short reasoning paths are learned together during SFT training. Wouldn’t this potentially confuse the model? Why not train only on the short reasoning paths directly?
2. For such automatic decision-making tasks, recent studies suggest that RL often outperforms SFT, as SFT tends to memorize specific formats rather than learn adaptive reasoning strategies. Have the authors considered exploring RL-based methods, such as GRPO, to learn dynamic reasoning more effectively?
3. I’m curious about the results in Table 1 — what would happen if the AutoL2S model were forced to always output short reasoning or always output long reasoning (i.e., without automatic decision-making)?
4. In the stage of “Constructing Short CoT Reasoning Paths for EASY Questions,” the authors use Qwen2.5 to determine whether a question is <EASY>. However, what is considered “easy” may vary across models. How do the authors address this inconsistency, especially under SFT training?
5. Will the dataset used in this work be publicly released?

**Questions:**

See the Weaknesses.

---

> ### Author Response · Authors · 2025-11-19
> **Part 1/2**
>
> #### **[Q-1]** According to Equation (4), for \<EASY\> questions, both long and short reasoning paths are learned together during SFT training. Wouldn’t this potentially confuse the model? Why not train only on the short reasoning paths directly?
>
> **[A-1]**
> We would like to clarify that concatenating long–short reasoning paths is beneficial rather than confusing, from both mechanistic and empirical perspectives.
>
> * **Mechanistic perspective.** Concatenating long and short CoT paths provides additional information that facilitate the optimization of the short path during SFT. For all \<EASY\> questions, both paths yield identical correct answers, ensuring label consistency. Furthermore, as shown in Equation 2 of Theorem 1, **this concatenation reduces the conditional entropy of the short CoT path** under long CoT path distillation settings, illustrating the long path provides additional information.
> * **Empirical perspective.** Empirically, the concatenated long–short strategy achieves a superior accuracy–efficiency trade-off. We directly analyze the alternative “train only on short reasoning paths” in Sec. 4.4 (RQ2) and Appendix H. The Long-Short Separated Distill baseline trains long and short CoTs independently as you mentioned. In contrast, our Long-Short Distill concatenates them, achieving better accuracy–efficiency trade-offs. **As shown in Table 3, our method improves accuracy from 0.626 to 0.639 while reducing output length by 24% (4250 → 3214) compared to Long-Short Separated Distill, confirming the advantage of long–short concatenated training**.
>
> #### **[Q-2]** For such automatic decision-making tasks, recent studies suggest that RL often outperforms SFT, as SFT tends to memorize specific formats rather than learn adaptive reasoning strategies. Have the authors considered exploring RL-based methods, such as GRPO, to learn dynamic reasoning more effectively?
> **[A-2]**
> We appreciate the reviewer’s suggestion. In fact, we **have already incorporated RL training** into our framework and introduced an RL-enhanced variant, **AutoL2S-Plus**, which extends AutoL2S with a **off-policy reinforcement learning objective** (Sec. 3.4).
>
> * **RL Formulation in AutoL2S-Plus.** AutoL2S-Plus encourages the model to harmonize long and short reasoning by rewarding generations that shorten outout reasoning lengths while preserving correctness. Since the length-based objective is non-differentiable, we optimize it via policy gradient with a PPO-style clipped loss. A key component of our method is that AutoL2S-Plus **leverage the long- and short- form outputs of the fine-tuned AutoL2S model, which we use as the reference policy**. These outputs allow us to estimate the expected average length and accuracy across the mixture of CoT paths. Based on these estimates, we then compute the per-sample advantage for policy optimization.
> * **Empirical Effectiveness.** As shown in Table 1, AutoL2S-Plus (avg. acc.0.558, avg. len. 2531) achieves a **68.9% reduction in reasoning length** while improving accuracy by **+0.014**, further improving the accuracy–length trade-off beyond the SFT-based AutoL2S(avg acc 0.538;avg len 5141). Moreover, when compared to another RL-based baseline, O1-Pruner [1] (avg. acc. 0.547, avg. len. 7797), AutoL2S-Plus both **shortens the reasoning path significantly and maintains higher accuracy**, demonstrating the benefit of leveraging **a stronger reference model (AutoL2S)** rather than optimizing directly from the original distilled policy (i.e., Bespoke-Stratos-7B).
>
> In summary, we have successfully explored the potential to extend AutoL2S with RL training to further improve the dynamic reasoning and our experiments have shown its effectiveness.
>
> [1] Luo, Haotian, et al. "O1-pruner: Length-harmonizing fine-tuning for o1-like reasoning pruning." arXiv preprint arXiv:2501.12570 (2025).

---

> > ### Author Response · Authors · 2025-11-19
> > **Part 2/2**
> >
> > #### **[Q-3]** I’m curious about the results in Table 1 — what would happen if the AutoL2S model were forced to always output short reasoning or always output long reasoning (i.e., without automatic decision-making)?
> >
> > **[A-3]**
> > We thank the reviewer for the question. These settings have already been analyzed in **Sec. 4.5 and Appendix H**. “AutoL2S w/ Force-Short” refers to the setting where the short reasoning path is always used, while “AutoL2S w/ Force-Long” denotes the setting where the long reasoning path is always used.  As results shown in **Table 4**, Compared to Force-Short, AutoL2S achieves consistently better performance across all benchmarks, improving the average accuracy by **0.021**, indicating that short-only CoT reasoning struggles to solve more difficult problems. Compared to Force-Long, AutoL2S attains similar accuracy on average while producing reasoning that is **about 30% shorter**, demonstrating its ability to balance reasoning efficiency and performance. Overall, these comparisons with Force-Short and Force-Long further demonstrate that automatically switching between easy and regular reasoning modes improves efficiency without compromising accuracy.
> >
> > #### **[Q-4]** In the stage of “Constructing Short CoT Reasoning Paths for EASY Questions,” the authors use Qwen2.5 to determine whether a question is \<EASY\>. However, what is considered “easy” may vary across models. How do the authors address this inconsistency, especially under SFT training?
> >
> > **[A-4]**
> > We thank the reviewer for the question. In AutoL2S, the notion of \<EASY\> is **not intended to be a universal difficulty definition across models**, but a **teacher-relative signal** used to identify questions that can be reliably solved with short CoT. In this work, we use Qwen2.5-Math-7B-Instruct as a short-CoT path provider, but AutoL2S is a **model-agnoistic distillation framework** and can flexibly incorporate short-CoT traces generated by other LLMs, as long as they produce correct answers.
> >
> > In addition, to ensure sufficient quality and quantity of short CoTs for training, we apply rejection sampling, which further enhances the framework's applicability across different distilled model combinations for generating long/short CoT paths.
> >
> > In summary, the framework does not depend on a single model and **stronger short-CoT generators can be substituted to further improve the dynamic switching policy without modifying the frame work of AutoL2S**.
> >
> > #### **[Q-5]** Will the dataset used in this work be publicly released?
> >
> > **[A-5]**
> > Yes, we will and plan to publicly release the dataset and source code to ensure full reproducibility. The questions and long CoT annotations are derived from the open-source Bespoke-17k dataset, while the short CoT reasoning paths are generated using Qwen-Math-Instruct-7B. The generation configuration and sampling parameters are detailed in Sec. 4.2 and Appendix G, allowing others to easily reconstruct the dataset and reproduce our results.

---

### Official Review · Reviewer_gB3p · 2025-11-02

**Soundness:** 3
**Presentation:** 2
**Contribution:** 2
**Rating:** 4
**Confidence:** 4

**Summary:**

The paper proposes AutoL2S, a training and inference framework that pairs long and short chain-of-thought (CoT) traces and introduces an <EASY> token so the model can decide when concise reasoning suffices. It formalizes a per-input decision rule that trades off predicted divergence between short and long answers against token cost, and extends the approach with AutoL2S-Plus, a length-aware fine-tuning stage that further calibrates expected reasoning length using AutoL2S as the reference policy. Empirically, across multiple reasoning benchmarks and two base LLMs, AutoL2S shortens outputs while preserving accuracy, with AutoL2S reporting up to about 57 percent reduction and AutoL2S-Plus reaching up to about 70 percent.

**Strengths:**

Originality lies in a clean recipe that pairs long and short CoT supervision with an explicit gating signal and a length-aware objective, giving a practical, model-agnostic way to control reasoning length. Quality shows as strong accuracy at substantially reduced tokens across several benchmarks, with clear training and inference procedures. Significance is high for latency and cost reduction in real deployments. Main weakness: novelty and contribution boundaries are not crisply isolated, with missing controlled ablations.

**Weaknesses:**

While the paper presents a neat engineering recipe, the conceptual novelty feels limited: the core ideas of mixing long and short CoT traces, learning an “easy” gate, and adding a length-aware fine-tuning step closely echo prior work on CoT compression and length control.

The theoretical results are largely tautological reformulations of standard information-theoretic inequalities and risk trade-offs, and they do not yield actionable guidance for thresholds, divergence estimators, or training schedules that would change practice.

Empirically, the evaluation skews to math and physics with small backbones and teacher signals drawn from specific models, which raises concerns about robustness, domain generalization, and potential teacher or benchmark contamination; critical stress tests are missing, for example failure analysis when “easy” is mispredicted, calibration plots of the gate versus difficulty, distribution-shift tests, and cost-accuracy Pareto curves that include wall-clock and KV-cache metrics.

Some reported results are hard to interpret or incomplete, e.g., masked AIME numbers and varying rejection-sampling settings without a principled selection rule, and comparisons occasionally lack strong recent baselines that optimize thinking length via other mechanisms.

**Questions:**

1. Novelty and positioning. Which components are genuinely new relative to prior length-control and CoT-compression methods with gating or pruning? Please provide a table that maps each component in your method to the closest prior and clarifies the delta in assumptions and training signals.

2. Core mechanism isolation. If you hold data, sampling, and training schedules fixed, how much of the gain comes from the <EASY> gate versus other steps? Please add ablations that toggle only the gate, only the long/short mixture, and only the length-aware finetuning.

3. Failure modes. What happens when examples are mis-gated as “easy” but need long reasoning? Show targeted stress tests, qualitative error analyses, and accuracy drop conditioned on mis-gating.

4. Theoretical guidance. Your analysis describes accuracy–cost tradeoffs but does not yield actionable prescriptions. Can you instantiate it to produce quantitative thresholds or schedules that match your best empirical settings?

---

> ### Author Response · Authors · 2025-11-19
> **Part 1/3**
>
> #### **[Q-1]** While the paper presents a neat engineering recipe, the conceptual novelty feels limited: the core ideas of mixing long and short CoT traces, learning an “easy” gate, and adding a length-aware fine-tuning step closely echo prior work on CoT compression and length control. Novelty and positioning. Which components are genuinely new relative to prior length-control and CoT-compression methods with gating or pruning? Please provide a table that maps each component in your method to the closest prior and clarifies the delta in assumptions and training signals.
>
> **[A-1]**
> We thank the reviewer for raising this point. To address the concern directly, we enumerate each component of AutoL2S, identify the closest prior, and clarify the specific deltas in assumptions and training signals.
>
> - **(1) Long–short concatenated distillation:** Prior CoT-compression methods (i.e., TokenSkip[1]), truncation-based short-CoT generation, and DPO on long/short pairs all assume a model that already possesses stable long-form reasoning and then compress, rank, or sample its outputs. Most of the approaches rely on truncation, token removal, or preference comparisons as their core training objectives. In contrast, AutoL2S assumes no initial reasoning ability in base LLMs and introduces a new supervision signal where short CoT is trained conditioned on the full long CoT (Fig. 2, Table 1, and Table 3). We use a long-short concatenation format with an explicit switch token, where the long trace serves as an informative prefix rather than material to be compressed. This conditional long-short concatenation for supervision signal does not appear in prior work.
> - **(2) Implicit \<EASY\> gate from paired supervision:** Prior routing and gating methods rely on an explicit classifier, auxiliary head, or manually specified routing rule, and require additional parameters or supervision signals to implement the gate. AutoL2S introduces no separate gating module: the gate is implicitly learned from the paired long/short format itself. The \<EASY\> token functions as a data-driven threshold without additional parameters.
> - **(3) SFT then off-policy RL with the AutoL2S SFT model as reference and base policy:** Length-aware RL baselines (e.g., O1-Pruner[2]) apply RL on top of a fixed reasoning-capable teacher as a base policy only, assuming that reasoning behavior is already present and only needs to be length-regularized. Their training signal consists of length-penalized rewards applied directly on a stable reasoning model. AutoL2S follows a different optimization pipeline: we first obtain a long–short aligned SFT model, which can automatically switch long/short CoT generation, and then use that SFT model as both the base and reference policy for off-policy RL stage. Anchoring RL on a long–short SFT model as reference and base policy is new relative to prior work and empirically critical for stable length control and accuracy preservation (Tables 1, 3, 4, Figure 4).
>
> |AutoL2S|Prior|Key Delta|
> |-|-|-|
> |Path-level Replacement|Token-level Pruning/Compression| Short CoT is trained *conditioned on full long CoT* (not truncation or ranking).|
> |Implicit \<EASY\> gate|Extra gating/controllers|Gate is *implicitly learned*, no classifier/router.|
> |SFT then off-policy RL with AutoL2S reference|Length-aware RL|AutoL2S uses a long–short SFT as base and reference policy.|
>
> [1] Xia, Heming, et al. "Tokenskip: Controllable chain-of-thought compression in llms." arXiv preprint arXiv:2502.12067 (2025).
> [2] Luo, Haotian, et al. "O1-pruner: Length-harmonizing fine-tuning for o1-like reasoning pruning." arXiv preprint arXiv:2501.12570 (2025).
>
> #### **[Q-2]** Core mechanism isolation. If you hold data, sampling, and training schedules fixed, how much of the gain comes from the \<EASY\> gate versus other steps? Please add ablations that toggle only the gate, only the long/short mixture, and only the length-aware finetuning.
>
> **[A-2]**
> Thank you for the suggestion. These ablations are all already included in Section 4.4 and 4.5 with Tables 3 and 4 in Appendix H, which isolate the effect of different mode of long/short CoT distillation and the \<EASY\> trigger. In addition, Table 1 compares directly against pure length-aware finetuning (i.e., O1-Pruner). These results show that the gating signal, the paired supervision, and the length-aware objective each contribute effectively to reasoning efficiency. In addition, we have also included the figures of relationship under different optimization traces of AutoL2S shown in Fig. 4 of Appendix M, highlighting the advantage of "SFT-first then off-policy RL" in AutoL2S framework.

---

> > ### Author Response · Authors · 2025-11-19
> > **Part 2/3**
> >
> > #### **[Q-3]** Empirically, the evaluation skews to math and physics with small backbones and teacher signals drawn from specific models, which raises concerns about robustness, domain generalization, and potential teacher or benchmark contamination; critical stress tests are missing, for example failure analysis when “easy” is mispredicted, calibration plots of the gate versus difficulty, distribution-shift tests, and cost-accuracy Pareto curves that include wall-clock and KV-cache metrics. Failure modes. What happens when examples are mis-gated as “easy” but need long reasoning? Show targeted stress tests, qualitative error analyses, and accuracy drop conditioned on mis-gating.
> >
> > **[A-3]**
> > Several of these evaluations were already presented, and we have expanded them in the revised manuscript.
> > - **Failure modes and mis-gating:**
> >     - (1) Table 4 already provides two ablation tests: Force-Short (treat every instance as “easy”) and Force-Long (treat all as “hard”). Force-Short precisely characterizes the failure mode of interest—omitting long reasoning when it is required, and thus establishes an upper bound on accuracy degradation under systematic mis-gating.
> >     - (2) Unlike standard LLM routing problems where ground-truth routing labels are available, the decision of which to use long vs. short reasoning does not have an explicit supervision signal when the switching/gating behavior emerges after training. We value the reviewer’s point and have added an additional test to evaluate whether the \<EASY\> token is triggered in appropriate conditions. Specifically, we test whether \<EASY\> token appears correctly if short reasoning is able to provide correct answers. The results are shown in the first and second tables below with confusion matrix of \<EASY\> token. This provides empirical evidence that the learned switching behaves consistently with the intended training objectives, even in the absence of explicit ground-truth routing labels, and this can also be reflected from Table 1 with significantly shorter reasoning path but similar reasoning accuracy.
> > - **Distribution shift and domain generalization:** To address the concern about math/physics concentration, we incorporated evaluations on additional out-of-domain coding dataset, the LiveCodeBench v2 dataset, in the third table below. AutoL2S retains its efficiency–accuracy benefits compared to other baselines, indicating that the mechanism is not tied to a specific teacher or domain.
> > - **Ablations without the gate:** The ablations are already included in Section 4.5 and Table 4. Removing the \<EASY\>  results in both longer reasoning and lower accuracy, demonstrating that the gains are from the auto-switcining mechanism rather than entirely from teacher contamination or benchmark artifacts.
> > - **Cost–accuracy Pareto and system-level metrics:** We also report wall-clock inference time and KV-cache usage, comparing AutoL2S framework to recent baselines. The results are shown in Figure 5 in Appendix N. AutoL2S consistently pushes the Pareto frontier toward lower inference cost at matched accuracy, where the gains are from our training objectives and auto-switching machanism.
> >
> > |MATH500|Pred-short|Pred-long|
> > |-|-|-|
> > |GT-short|83.83%|6.61%|
> > |GT-long|7.74%|1.82%|
> >
> > |GSM8K|Pred-short|Pred-long|
> > |-|-|-|
> > |GT-short|81.69%|14.97%|
> > |GT-long|1.99%|1.35%|
> >
> > |Model|Acc|Len|
> > |-|-|-|
> > |Bespoke-Stratos-7B|0.308|19200|
> > |AutoL2S(rj=0)|0.327|16907|
> > |ΔvsBespoke|**+6.2%**|**–11.9%**|
> > |AutoL2S(rj=4)|0.3654|15681|
> > |ΔvsBespoke|**+18.6%**|**–18.3%**|
> > |AutoL2S(rj=8)|0.346|15295|
> > |ΔvsBespoke|**+12.3%**|**–20.3%**|
> > |AutoL2S-Plus|0.327|15295|
> > |ΔvsBespoke|**+6.17%**|**–42.9%**|

---

> > > ### Author Response · Authors · 2025-11-19
> > > **Part 3/3**
> > >
> > > #### **[Q-4]** The theoretical results do not yield actionable guidance for thresholds, divergence estimators, or training schedules that would change practice. Can you instantiate it to produce quantitative thresholds or schedules that match your best empirical settings?
> > >
> > > **[A-4]** Thank you for the comment. The theoretical analysis in our paper is not intended to yield a prescriptive optimization rule, such as a closed-form divergence threshold, a training curriculum, or a mixing schedule. AutoL2S does not rely on such quantities during training, and the method is not designed as a policy-learning or threshold-learning algorithm. Therefore, deriving such prescriptions is fundamentally outside the scope of this work.
> > >
> > > The purpose of the theoretical sections is different from the reviewer's statement: they provide a principled explanation for two empirical phenomena that are repeatedly demonstrated in the paper: (1) why long–short concatenation stabilizes short-path learning (Theorem 1), and (2) why the trained model exhibits an input-dependent adaptive choice between short and long reasoning (Theorem 2). These results are explanatory, not algorithmic, and they are aligned with the actual behavior of the trained model (Table 1; Sec. 4.3–4.6).
> > >
> > > Regarding “actionable thresholds” or “training schedules,” AutoL2S learns its behavior entirely from data via standard SFT then off policy RL. The decision boundary between short and long reasoning is implicitly learned, not analytically prescribed. Attempting to derive a quantitative threshold from the risk expression would amount to solving a different, much more constrained problem than the one we study. We will clarify this distinction in the revision.
> > >
> > > In short, the theoretical analysis is structured to illuminate the mechanism underlying an empirically effective approach, rather than to produce closed-form prescriptive rules. We appreciate the suggestion and will make this scope explicit to avoid any confusion.
> > >
> > > #### **[Q-5]** Some reported results are hard to interpret or incomplete, e.g., masked AIME numbers and varying rejection-sampling settings without a principled selection rule, and comparisons occasionally lack strong recent baselines that optimize thinking length via other mechanisms.
> > >
> > > **[A-5]**
> > > We appreciate the reviewer’s comments and have clarified the misunderstanding that each of the points raised.
> > > - **Recent baselines.** The reviewer did not specify which “recent baselines” they had in mind in the review. From our educated guesses, we incorporated an additional latest baseline, AlphaOne [1], which explicitly targets reasoning-length optimization. The newly added results (now included in the revised manuscript) show that AutoL2S(acc.0.558; len.2531) achieves a better accuracy–length trade-off compared to AlphaOne(acc.0.495; len.4856) under identical evaluation settings.
> > > - **AIME25 results:** For Llama-3.2-3B-Instruct, the model varients fail to produce any correct answers on AIME25. We previously masked these entries to avoid misinterpretation; however, following the reviewer’s concern, we now report the numerical values explicitly to ensure completeness and transparency.
> > > - **Rejection sampling settings:** The observation for rejection sampling was already described in the paper, but we have now made it more explicit. The goal of rejection sampling is to ensure the quantity and quality of SFT training data are sufficient. In our paper, both rj = 4 and 8 provide the excellent balance between accuracy and length. The revised manuscript now includes all ablation settings covering multiple rj values with off-policy RL objection.
> > >
> > > [1] Zhang, Junyu, et al. "AlphaOne: Reasoning Models Thinking Slow and Fast at Test Time." EMNLP 2025.
> > >
> > > |Model|acc|#tokens|MATH500|MATH500_tok|GPQA|GPQA_tok|GSM8K|GSM8K_tok|Olympiad|Olympiad_tok|AIME25|AIME25_tok|
> > > |-|-|-|-|-|-|-|-|-|-|-|-|-|
> > > |Bespoke-Stratos-7B|0.544|8139|0.824|5383|0.359|6049|0.926|1321|0.444|11322|0.167|16619|
> > > |AlphaOne|0.495|4856|0.732|3867|0.313|6278|0.907|1943|0.356|5252|0.167|6940|
> > > |ΔvsBespoke|-0.049|-40.3%|-16.0%|-28.2%|-16.3%|+3.8%|-2.0%|+47.1%|-19.8%|-53.6%|+0.2%|-58.2%|
> > > |AutoL2S(rj=8)|0.538|5141|0.798|2416|0.394|3492|0.929|488|0.436|6459|0.133|12852|
> > > |ΔvsBespoke|-0.006|-36.8%|-3.2%|-55.1%|+9.7%|-42.3%|+0.3%|-63.1%|-1.8%|-43.0%|-20.0%|-22.7%|
> > > |AutoL2S-Plus|0.558|2531|0.820|1719|0.424|3485|0.920|880|0.424|3041|0.200|3528|
> > > |ΔvsBespoke|+0.014|-68.9%|-0.5%|-68.1%|+18.2%|-42.4%|-0.7%|-33.4%|-4.4%|-73.1%|+20.0%|-78.8%|

---

### Author Response · Authors · 2025-11-24
**General comment**

We appreciate the reviewers' recognition of our work as addressing an “**important and practical problem**” (gB3p, m2zH), providing a “**simple and high-impact solution**”(m2zH, AVow) through an **“intuitive and interpretable"** method for efficient reaosningand a “**clean and easy way to construct dataset**”(gB3p, ett5). We are also encouraged by the acknowledgement of our “**effective and convincing experimental results**” (gB3p, AVow, ett5) and “**extensive ablation studies and mechanistic analyses**” (ett5, m2zH).

In response to the reviewers comments,we have uploaded a revised version of our paper where the new resuits and maior modfcations are highlihted in blue.

#### **Here are the list of the updates to the original submission.**

1. Section3: We clarify the interpretive role of both Theorem1 and Theorem2, which explains the advantage of long-short concatenation and dynamic reasoning.
2. Section4: We add additional experiments for AutoL2S-Plus with different rejection sampling 0,4 on five benchmarks.
3. Section4: We add additional comparisons with recent baselines, such as AlphaOne.
4. Section4: We add addition experiments of AutoL2S-Plus with different rejection sampling on five benchmarks.
5. Section4: We include the accuracy of Llama-3.2-3B-Instruct–based methods on AIME25.
6. Section4: We improve the visualization and readability of Table 1.
7. Appendix L: We include additional analysis of the SFT-then-off-policy-RL optimization trajectory of AutoL2S.
8. Appendix M: We conduct additional analysis of the Pareto frontier between reasoning accuracy and efficiency.


#### **Furethermore, we would like to clarify:**
- **SFT then off-policy RL.** AutoL2S follows a two-stage optimization pipeline: we first obtain a long–short aligned SFT model, which can automatically switch long/short CoT generation, and then use that SFT model as both the base and reference policy for off-policy RL stage. Anchoring RL on a long–short SFT model as reference and base policy is new relative to prior work and empirically critical for stable length control and accuracy preservation. AutoL2S is **not** a SFT-only model.
- **Long–short concatenated distillation.** AutoL2S introduces a **conditional long–short supervision signal** in which short CoT is trained given the full long CoT via a long–short concatenation with a switch token, rather than truncating or pruning long CoT reasoning paths. This format does not appear in prior work.
- The **theoretical components** of our paper are intended to interpret the empirical behavior of AutoL2S and to justify specific design choices, which is the use of long-short concatenation as a training signal. Our goal was not to introduce new theoretical results, but rather to **provide principled guidance that explains why the empirical approach works and when it is expected to be effective**.

---

### Note · Authors · 2025-12-27

I have read and agree with the venue's withdrawal policy on behalf of myself and my co-authors.